# Similarity-Distance-Magnitude Activations

## Abstract

We introduce the SIMILARITY-DISTANCE-MAGNITUDE (SDM) activation function, a more robust and interpretable formulation of the standard softmax activation function, adding SIMILARITY (i.e., correctly predicted depth-matches into training) awareness and DISTANCE-to-training-distribution awareness to the existing output MAGNITUDE (i.e., decision-boundary) awareness, and enabling interpretability-by-exemplar via dense matching. We further introduce the SDM estimator, based on a data-driven partitioning of the class-wise empirical CDFs via the SDM activation, to control the class- and prediction-conditional accuracy among selective classifications. When used as the final-layer activation over pre-trained language models for selective classification, the SDM estimator is more robust to co-variate shifts and out-of-distribution inputs than existing calibration methods using softmax activations, while remaining informative over in-distribution data.

## 1 Introduction

Neural-network-based language models (LMs) pose a challenge for interpretable and reliable deployment given the non-identifiability of their parameters (Hwang & Ding, 1997, inter alia)[1], which can number in the billions or more. Instead of directly interpreting parameters, one option is to move the focus of interpretation to auditing predictions as a form of interpretability by example, or *exemplar*, over the representation space of such models via dense matching. However, for real-world deployments, robust approaches for predictive uncertainty are also needed, both for human decision-making and for constructing sequentially dependent LM pipelines.

Known theoretical results limit the statistical quantities that can be derived over LMs. Statistical guarantees in the distribution-free setting are limited to approximately conditional quantities (Valiant, 1984; Lei & Wasserman, 2014; Foygel Barber et al., 2020, inter alia). Further, even typical approximately conditional quantities can be difficult to obtain in practice, since the minimal assumption of exchangeability with a known held-out data set is itself often violated with co-variate and label shifts, which can be difficult to foresee with existing methods. Epistemologically, the prevalence of hallucinations and highly-confident wrong answers with widely deployed LMs suggests a technical impasse in effectively modeling the predictive uncertainty, despite significant work from Bayesian, Frequentist, and empirically motivated perspectives (Gal & Ghahramani, 2016; Angelopoulos et al., 2021; Guo et al., 2017; Lakshminarayanan et al., 2017; Ovadia et al., 2019, inter alia). A foundational piece is evidently missing from the picture.

Given these intrinsic challenges, we approach the problem of uncertainty quantification over LMs from a new angle and ask: *Can we leverage the metric learning and dense matching capabilities of neural networks over high-dimensional inputs to at least aim to maximize, with minimal distributional assumptions, the separation of aleatoric (irreducible) uncertainty and epistemic (reducible) uncertainty, decomposing the sources of the latter in a manner that is interpretable and actionable?*

We answer this question in the affirmative with a conceptually parsimonious, data-driven partitioning of the data to decompose sources of epistemic uncertainty: Correctly predicted depth-matches into the training set

---

[1]Informally, this means that two or more distinct sets of values for the parameters can result in identical output distributions. As a consequence, interpreting the parameters of such models is typically much more complicated than with a simple linear regression model, for example.

(SIMILARITY), the DISTANCE to the training set, and the distance to the decision-boundary (MAGNITUDE). We use these signals to construct a new activation function, the SDM activation, which can be used as a replacement for the standard softmax activation, as, for example, the final-layer activation. The SDM activation enables more reliable estimates of the predictive uncertainty for selective classification (Chow, 1957; Geifman & El-Yaniv, 2017, inter alia), which addresses the need for uncertainty quantification with LMs used in multi-stage decision pipelines, in settings subject to co-variate shifts and out-of-distribution inputs.

In summary, in this work:

- We introduce the SIMILARITY-DISTANCE-MAGNITUDE (SDM) activation function, which encodes strong signals of epistemic uncertainty, for use with neural networks.

- We introduce the SDM estimator for use in controlling class- and prediction-conditional accuracy among selective classifications, based on a data-driven partitioning of the class-wise empirical cumulative distribution functions (eCDFs) over the output via the SDM activation.

- We examine the behavior of the SDM activation as a final-layer activation over pre-trained language models, using the SDM estimator for selective classification. We demonstrate empirically that the SDM estimator is more robust to co-variate-shifts and out-of-distribution inputs than existing classes of post-hoc calibration methods over softmax activations, while remaining informative over in-distribution data.

## 2 Preliminaries

### 2.1 Setting

We consider the standard multi-class classification setting (e.g., predicting the sentiment of a movie review). We are given a training dataset, $\mathcal{D}_{\text{tr}} = \{(\boldsymbol{x}_n, y_n)\}_{n=1}^N$ of inputs, $\boldsymbol{x} \in \mathcal{X}$, paired with their corresponding ground-truth discrete labels, $y \in \mathcal{Y} = \{1, \ldots, C\}$, and a labeled calibration dataset, $\mathcal{D}_{\text{ca}}$, drawn from the same distribution as $\mathcal{D}_{\text{tr}}$. We are then given a new test instance, $\boldsymbol{x}$, from an unlabeled test set, $\mathcal{D}_{\text{te}}$, and seek to estimate the label with a prediction, $\hat{y}$, via the un-normalized log probabilities ("logits", informally) of a final linear layer: $\boldsymbol{z} = \boldsymbol{W}^T \boldsymbol{h} + \boldsymbol{b}$, where $\boldsymbol{h} = \text{network}(\boldsymbol{x}; \theta)$ is the final hidden state of a network parameterized by $\theta$. The discrete prediction is taken as $\hat{y} = \arg\max \boldsymbol{z}$; however, for learning $\theta$, $\boldsymbol{W}$, and $\boldsymbol{b}$, and for human decision-making, we also seek an estimate of the predictive uncertainty, $p(y \mid \boldsymbol{x})$, which is typically obtained by normalizing $\boldsymbol{z}$ via the softmax activation described next. We will make a distinction between models, $\mathcal{M}$ (defined by $\theta$, $\boldsymbol{W}$, and $\boldsymbol{b}$, and when applicable, the exemplar adaptor, described below), which produce the prediction, $\hat{y}$, and estimators, $\mathcal{E}$, which provide an estimate of $p(y \mid \boldsymbol{x})$, because different estimators can be used over the same model.

### 2.2 Softmax and the Cross-Entropy loss

The softmax activation is commonly used in neural network architectures, including, for example, as a router in self-attention mechanisms (Vaswani et al., 2017) and mixture-of-experts models (Shazeer et al., 2017), and forming the basis of the cross-entropy loss used for next-token training of LMs. It is the typical final output layer of LMs, converting the un-normalized model logits to a normalized probability distribution:

$$\text{softmax}(\boldsymbol{z})_i = \frac{e^{\tau \cdot z_i}}{\sum_{c=1}^C e^{\tau \cdot z_c}}, 1 \leq i \leq C, \tau \geq 0 \tag{1}$$

The inverse-temperature parameter, $\tau$, controls the sharpness of the distribution. As $\tau \to 0$, the output of $\text{softmax}(\boldsymbol{z})$ converges to a uniform distribution where each class has probability $\frac{1}{C}$; as $\tau \to \infty$, the output converges to a distribution in which all of the mass is assigned to a single class. In deep learning, $\tau$ is treated as a learnable, *global* hyper-parameter; *instance-wise* variation in the distance to the decision-boundary is thus determined by the relative MAGNITUDE of $z_{\hat{y}}$. This model is learned by minimizing the cross-entropy loss between $\boldsymbol{z}$ and the index of the true labels over $\mathcal{D}_{\text{tr}}$. The *natural* logarithm of the loss is the counterpart

to the base $e$ of the softmax:

$$\mathcal{L}(\theta, \boldsymbol{W}, \boldsymbol{b}; \mathcal{D}_{\text{tr}}) = -\frac{1}{N} \sum_{n}^{N} \log_e \left( \frac{e^{\tau \cdot z_{y_n}}}{\sum_{c=1}^{C} e^{\tau \cdot z_c}} \right) \tag{2}$$

## 3 Methods

In this work, we revisit Eq. 1 and 2. We seek to decouple the sources of epistemic uncertainty via a new activation function that is conceptually:

$$\text{SDM}(\boldsymbol{z})_i = \frac{\text{SIMILARITY}^{\text{DISTANCE} \cdot \text{MAGNITUDE}_i}}{\sum_{c=1}^{C} \text{SIMILARITY}^{\text{DISTANCE} \cdot \text{MAGNITUDE}_c}} \tag{3}$$

with a corresponding negative log likelihood loss that takes into account the change of base (§ 3.1). Unique to this setting, a modification to label-conditional conformal prediction (Vovk et al., 2005) then follows via a parsimonious partitioning of the class-wise empirical CDFs, providing a principled basis for controlling the class-conditional accuracy among selective classifications, combined with empirically-robust prediction-conditional estimates.

### 3.1 Similarity-Distance-Magnitude Activation Functions

Calculating the SDM activation involves training an exemplar adaptor, a 1-D CNN adaptor (with a final linear layer) over the frozen hidden states of a network, to induce distilled, compressed representations of the underlying network's representation space conditional on its predictions. The resulting representations provide a probabilistic mapping to the training, or support, set. In this way, neural networks, including large pre-trained networks, can be viewed as *hidden* instance-based metric learners, from which we can then derive signals of the epistemic uncertainty.

#### 3.1.1 Exemplar Adaptor

We take as the CNN of our exemplar adaptor $g : \boldsymbol{h} \in \mathbb{R}^D \mapsto \boldsymbol{h}' \in \mathbb{R}^M$, a 1-D CNN that takes as input $\boldsymbol{h}$ of the underlying network. The CNN has $M$ filters, the filter applications of which produce $\boldsymbol{h}'$, the distilled representation of the underlying network. A final linear layer, $\boldsymbol{z}' = \boldsymbol{W}'^T \boldsymbol{h}' + \boldsymbol{b}', \boldsymbol{z}' \in \mathbb{R}^C$, then replaces the underlying network's linear layer, with the discrete prediction taken as $\hat{y} = \arg\max \boldsymbol{z}'$. This exemplar adaptor will then enable us to derive the SIMILARITY, DISTANCE, and MAGNITUDE values, as defined next.

#### 3.1.2 Similarity

We define the SIMILARITY ($q$) of an instance to the training set as the count of consecutive nearest matches in $\mathcal{D}_{\text{tr}}$ that are correctly predicted *and* match $\hat{y}$ of the test instance.[2] Concretely, we first sort $\mathcal{D}_{\text{tr}}$ (for which we have both model predictions and ground-truth labels) based on the $L^2$ distance from $\boldsymbol{h}'$, $\left[ (\boldsymbol{x}_{(1)}^{tr}, \hat{y}_{(1)}^{tr}, y_{(1)}^{tr}), \dots, (\boldsymbol{x}_{(N)}^{tr}, \hat{y}_{(N)}^{tr}, y_{(N)}^{tr}) \right]$, such that $||\boldsymbol{h}' - \boldsymbol{h}_{(1)}'^{tr}||_2 \leq \dots \leq ||\boldsymbol{h}' - \boldsymbol{h}_{(N)}'^{tr}||_2$, and then calculate $q \in \{0, \dots, |\mathcal{D}_{\text{tr}}|\}$ as:

$$q = \sum_{i=1}^{|\mathcal{D}_{\text{tr}}|} \mathbf{1}_{\hat{y} = \hat{y}_{(i)}^{\text{tr}}} \cdot \mathbf{1}_{\hat{y}_{(i)}^{\text{tr}} = y_{(i)}^{\text{tr}}} \cdot \mathbf{1}_{i-1 = \sum_{j=1}^{i-1} \mathbf{1}_{\hat{y} = \hat{y}_{(j)}^{\text{tr}} \cdot \mathbf{1}_{\hat{y}_{(j)}^{\text{tr}} = y_{(j)}^{\text{tr}}}} \tag{4}$$

where the rightmost indicator function, $\mathbf{1} \in \{0, 1\}$, ensures consecutive (depth-wise) matches.[3] By definition, $q$ cannot exceed the count of the most prevalent class label in $\mathcal{D}_{\text{tr}}$, and since we assume an approximately equal number of points for each class, $q \ll |\mathcal{D}_{\text{tr}}|$ is typical. For the special case of calculating $q$ for $\boldsymbol{x} \in \mathcal{D}_{\text{tr}}$, which only occurs during learning, we exclude the self-match.

---

[2]We use the letter $q$, as this value **q**uantizes the closeness of a point to the training set with a discrete estimate.

[3]This seemingly simple rule differs from traditional KNN rules (Cover & Hart, 1967; Devroye et al., 1996, inter alia) in two critical respects: The neural network serves as a semi-supervised metric learner of the distances between the dense representations that identify the instances, and there is a model prediction (in addition to the ground-truth label) for each instance in the support set. The former enables effective partitioning despite the curse of high dimensions; the latter provides an additional indicator of reliability for each instance.

### 3.1.3 Distance

The $L^2$ distance to the nearest match in $\mathcal{D}_{\mathrm{tr}}$ follows from above: $d_{\mathrm{nearest}} = ||\boldsymbol{h}' - \boldsymbol{h}_{(1)}'^{tr}||_2$. We normalize these values by defining the DISTANCE, $d \in [0, 1]$, in terms of the class-wise empirical CDFs of $d_{\mathrm{nearest}}$ over $\mathcal{D}_{\mathrm{ca}}$, as the most conservative quantile relative to the distance to the nearest matches observed in the labeled, held-out set:

$$d = \min\left[1 - \mathrm{eCDF}_{\mathrm{ca}}^{y_1}(d_{\mathrm{nearest}}), \ldots, 1 - \mathrm{eCDF}_{\mathrm{ca}}^{y_C}(d_{\mathrm{nearest}})\right] \tag{5}$$

The empirical CDFs are determined by the labeled points in $\mathcal{D}_{\mathrm{ca}}$ for which $q > 0$, where, as indicated by the superscripts, the stratification of points is by the true labels, $y$. For example, $\mathrm{eCDF}_{\mathrm{ca}}^{y_1}(d_{\mathrm{nearest}})$ is the empirical CDF of $d_{\mathrm{nearest}}$ values in $\mathcal{D}_{\mathrm{ca}}$ for which $y = 1$, a notation convention we will use throughout. (Points with $q = 0$ are effectively out-of-distribution points and treated as such in downstream decision-making, so they are excluded to avoid biasing the estimates.) At test time, we do not see $y$; instead, the minimum is calculated over the quantiles of each of the class-conditional eCDFs, regardless of $\hat{y}$. As with $q$, for the special case of calculating $d$ for $\boldsymbol{x} \in \mathcal{D}_{\mathrm{tr}}$, we replace $\mathrm{eCDF}_{\mathrm{ca}}^{y_c}$ with the analogous $\mathrm{eCDF}_{\mathrm{tr}}^{y_c}$, the class-wise empirical CDFs of $d_{\mathrm{nearest}}$ over $\mathcal{D}_{\mathrm{tr}}$ excluding self-matches.

### 3.1.4 Magnitude

We take as the MAGNITUDE, or distance to the decision boundary, $z_{\hat{y}}'$, as in the standard softmax case but via $\boldsymbol{z}'$ from the linear layer of the exemplar adaptor.

### 3.1.5 SDM Activation: Formulation

We use the above quantities to define the SDM activation function:

$$\mathrm{SDM}(\boldsymbol{z}')_i = \frac{(2+q)^{d \cdot z_i'}}{\sum_{c=1}^{C}(2+q)^{d \cdot z_c'}}, 1 \le i \le C \tag{6}$$

The output distribution becomes sharper with higher values of $q$, $d$, and $z'$. When $d_{\mathrm{nearest}}$ exceeds the largest distance observed in the labeled data, $d = 0$ and the output distribution is uniform, reflecting maximally high uncertainty. The standard softmax with $\tau = 1$ is recovered by setting $q = e - 2, d = 1$. As with the softmax activation, $\arg\max \mathrm{SDM}(\boldsymbol{z}') = \arg\max \boldsymbol{z}'$.

### 3.1.6 SDM Activation: Loss and Training

A loss analogous to Eq. 2 then follows with the applicable change of base. We use this loss to train the weights of the exemplar adaptor, which includes the parameters of the linear layer ($\boldsymbol{W}'$ and $\boldsymbol{b}'$), as well as the convolution weights and biases, which we collectively represent with $\boldsymbol{G}$. The weights of the underlying network remain fixed.[4]

$$\mathcal{L}(\boldsymbol{G}, \boldsymbol{W}', \boldsymbol{b}'; \mathcal{D}_{\mathrm{tr}}) = -\frac{1}{N}\sum_{n}^{N}\log_{(2+q)}\left(\frac{(2+q)^{d \cdot z_{y_n}'}}{\sum_{c=1}^{C}(2+q)^{d \cdot z_c'}}\right) \tag{7}$$

The first epoch of training is initialized with a standard softmax (i.e., setting $q = e - 2, d = 1$). Training then proceeds by re-calculating $q$ and $d$ for each $\boldsymbol{x} \in \mathcal{D}_{\mathrm{tr}}$ after each epoch. We take as the stopping criteria for one learning round as the epoch with the lowest balanced (across classes) average loss over $\mathcal{D}_{\mathrm{ca}}$. We repeat this process for $J$ iterations of random shuffles and splits of $\mathcal{D}_{\mathrm{tr}}$ and $\mathcal{D}_{\mathrm{ca}}$ and parameter initializations, choosing the final model as that with the globally lowest balanced (across classes) average loss over $\mathcal{D}_{\mathrm{ca}}$.

## 3.2 Evaluating Selective Classification

As a common, unambiguous baseline quantity for comparing selective classifiers over a held-out test set, $\mathcal{D}_{\mathrm{te}}$, we seek an easy-to-interpret and easy-to-evaluate metric, reflecting real-world applications. Among the

---

[4]As indicated by our loss notation and described in § 3.1.2 and § 3.1.3, instance-wise $q$ and $d$ are not parameters learned directly via gradient descent. By design, we are not back-propagating through a bi- and/or cross-encoded search graph.

selective classifications from an estimator, we seek (Quantity I) prediction-conditional accuracy at or above a given threshold, $\alpha \in (\frac{1}{C}, 1]$, and (Quantity II) class-conditional accuracy at or above that same threshold, $\alpha$.

To evaluate this metric, we only consider the points for which the given estimator assigns a high-probability of at least $\alpha$, which is typically near 1, such as $\alpha = 0.95$ in our experiments. We refer to this set of points as the *admitted*, or *non-rejected*, set. Then, given ground-truth values for $\mathcal{D}_{\text{te}}$, we assess whether the conditional accuracies of the admitted set are at least $\alpha$ when (Quantity I) stratifying by the predicted labels, $\hat{y}$, and when (Quantity II) stratifying by the true labels, $y$.

The estimator that rejects all points would meet these conditions. However, given two estimators that meet these conditions, we prefer that which rejects fewer points, ceteris paribus. In other words, we seek estimators that meet our reliability condition and are informative (i.e., maximize the number of points that are properly admitted), but when the estimator is uncertain, we prefer rejection over unexpectedly falling under the desired $\alpha$ probability threshold.

Quantity I corresponds to top-label calibration (Gupta & Ramdas, 2022), but with a single bin for evaluation, $[\alpha, 1]$, removing ambiguity with regard to the choice of binning the probabilities. Quantity II does not directly correspond to quantities typically examined in the calibration literature (Brier, 1950; Dawid, 1982; Guo et al., 2017; Vaicenavicius et al., 2019; Kull et al., 2019, inter alia), but it approximates[5] label-conditional conformal coverage in the special case of class-wise thresholds that only admit prediction sets of cardinality 1. We introduce a straightforward procedure to estimate this quantity next.

### 3.2.1 Controlling the Class-conditional Accuracy among Selective Classifications with SDM Estimators

In general, the statistical coverage guarantee of marginal split-conformal estimators is not directly informative for selection, since the coverage guarantee is not conditional on the set size. We may instead seek one of various approximately conditional notions of coverage (Romano et al., 2020; Angelopoulos et al., 2021, inter alia); however, there is no guarantee that when we stratify by sets of cardinality 1, coverage will be maintained. However, there is a *special case* in which label-conditional conformal estimators *do* provide a meaningful notion of class-conditional coverage for selection. Assuming $\mathcal{D}_{\text{ca}}$ and $\mathcal{D}_{\text{te}}$ are exchangeable, if the conformity score for each label is from a categorical distribution and the resulting thresholding of the class-wise empirical CDFs results in class-wise thresholds that are all greater than $\frac{1}{C}$, then the cardinality 1 sets will, on average, obtain class-conditional coverage, by definition.[6] Unfortunately, it may be rare to encounter this restricted setting over the full data distributions of real-world tasks. Instead, we will use the SDM activation to estimate a partitioning of the distribution into a region that approximately fulfills these assumptions.

First, we rescale the SIMILARITY estimate to take into account the DISTANCE and MAGNITUDE, given the predicted class. The resulting value[7] will be the basis for partitioning the distribution:

$$q' = \min\left(q, (2+q)^{\text{SDM}(\boldsymbol{z}')_{\hat{y}}}\right) \tag{8}$$

Next, we estimate label-conditional conformal thresholds, $[\psi_1, \ldots \psi_C]$, over the output from the SDM activation among a subset of the distribution constrained by progressively larger values of $q'$ (among $\lfloor q' \rfloor > 0$) until all thresholds are at least $\alpha$. By setting our stopping criteria at $\alpha$ rather than $\frac{1}{C}$, we also restrict the region to our empirically-motivated prediction-conditional quantity, $\text{SDM}(\boldsymbol{z}')_{\hat{y}}$. The procedure appears in Alg. 1. If we find a finite $q'_{\min}$ that obtains such thresholds, we refer to the resulting region as the HIGH-RELIABILITY

---

[5]For simplicity of presentation, we omit consideration of the Beta-distributed error term that is a function of the effective sample size of split-conformal coverage (Vovk, 2012). In practice, this term is negligible in the present experiments given $|\mathcal{D}_{\text{ca}}|$ and the resolution of the comparisons. See Appendix A.3 for a further discussion of analyzing the effective sample size for both the class-conditional and prediction-conditional estimates.

[6]Although the formal interpretations are not identical, the evaluation of class-conditional coverage for a single estimate of such cardinality 1 prediction sets in this restricted setting over $\mathcal{D}_{\text{te}}$ is numerically equivalent to assessing the class-conditional accuracy, when not considering the sample size error term.

[7]The min ensures that $q'$ remains 0 for points with $q = 0$, which are effectively out-of-distribution points.

($\text{SDM}_{\text{HR}}$) region, taking membership in this region as our selection criteria:

$$\text{SDM}_{\text{HR}} := \begin{cases} \hat{y} & \text{if } q' \geq q'_{\min} \wedge \text{SDM}(\boldsymbol{z}')_{\hat{y}} \geq \psi_{\hat{y}} \\ \bot & \text{otherwise} \end{cases} \tag{9}$$

where $\bot$ indicates a rejected (non-admitted) point and $\hat{y} = \arg\max \boldsymbol{z}'$.

To calculate this quantity for new, unseen test points $\boldsymbol{x} \in \mathcal{D}_{\text{te}}$, we require $\mathcal{D}_{\text{tr}}$ to calculate $q$ and $d_{\text{nearest}}$; the cached class-wise empirical CDFs of the distances over $\mathcal{D}_{\text{ca}}$ of Eq. 5; and $q'_{\min}$ and the thresholds, $[\psi_1, \ldots \psi_C]$. Evaluation of the $\text{SDM}_{\text{HR}}$ selection criteria is straightforward: We simply assess the conditional accuracies for the admitted points after stratifying by the predictions and the true labels, each in turn.

When Alg. 1 returns $q'_{\min} = \infty$, we obtain a useful empirical indicator that the model is too weak, or the data insufficient, to reliably obtain class- and prediction-conditional estimates at the specified $\alpha$ value.

---

**Algorithm 1** Search Algorithm to Find $q'_{\min}$ and $[\psi_1, \ldots \psi_C]$ to Estimate the HIGH-RELIABILITY Region

---
**Input:** cached $(q', \text{SDM}(\boldsymbol{z}')) \ \forall \ \boldsymbol{x} \in \mathcal{D}_{\text{ca}}, \ \alpha \in (\frac{1}{C}, 1]$
1: **procedure** ESTIMATE-HIGH-RELIABILITY-REGION(cached $(q', \text{SDM}(\boldsymbol{z}')) \ \forall \ \boldsymbol{x} \in \mathcal{D}_{\text{ca}}, \ \alpha \in (\frac{1}{C}, 1])$
2:      $q'_{\min} \leftarrow \infty$                                                     ▷ A finite $q'_{\min}$ may not be found.
3:      $[\psi_1, \ldots \psi_C] \leftarrow [\infty, \ldots, \infty]$                              ▷ Class-wise output thresholds
4:      sortedList $\leftarrow$ sorted $[q' \in \mathcal{D}_{\text{ca}} \text{ s.t. } \lfloor q' \rfloor > 0]$          ▷ Restricted to $\lfloor q' \rfloor > 0$ to exclude OOD
5:      **for** $q^* \in$ sortedList **do**
6:          Construct $\text{eCDF}_{\text{ca}}^{y_1}, \ldots, \text{eCDF}_{\text{ca}}^{y_C}$ for all $q' \geq q^*$ in $\mathcal{D}_{\text{ca}}$      ▷ eCDFs for $\text{SDM}(\boldsymbol{z}')$ (Eq. 6), stratified by $y$
7:          Calculate $\psi_c = \text{inverseCDF}_{\text{ca}}^{y_c}(1 - \alpha) \ \forall \ c \in \{1, \ldots, C\}$      ▷ Quantile functions are inverses of L. 6
8:          **if** all( $[\psi_1, \ldots \psi_C] \geq \alpha$ ) **then**                        ▷ Element-wise comparison
9:              $q'_{\min} \leftarrow q^*$
10:              **break**
11:      **return** $q'_{\min}, [\psi_1, \ldots \psi_C]$
**Output:** $q'_{\min}, [\psi_1, \ldots \psi_C]$

---

The convention is to refer to the basic architecture of Eq. 6 as the SDM activation function, and using the activation with the selection criteria of Eq. 9 as the SDM estimator.

## 4 Experiments

We provide controlled comparisons of our proposed methods over representative LMs and tasks, systematically ablating relevant components, holding the data and underlying LM constant, ceteris paribus. We consider in-distribution, co-variate shifted, and out-of-distribution test sets. We consider representative estimators over the existing LM architecture (i.e., without additional parameters); with CNN adaptors; and with the SDM activation layer.

### 4.1 Task: Binary Sentiment Classification

**Sentiment: $\mathcal{D}_{\text{tr}}$ and $\mathcal{D}_{\text{ca}}$.** Our first task is predicting the sentiment of movie reviews. We use the commonly used benchmark data of Maas et al. (2011). This is a binary classification task with $y \in \{0 = \text{negative}, 1 = \text{positive}\}$. $\mathcal{D}_{\text{tr}}$ and $\mathcal{D}_{\text{ca}}$ are constructed from a total of 18k instances. The held-out set for evaluation, $|\mathcal{D}_{\text{te}}| = 1583$, is from the same distribution as $\mathcal{D}_{\text{tr}}$ and $\mathcal{D}_{\text{ca}}$. This is a well-studied task for which the surface-level signals correlated with the target labels are expected to be effectively modeled by large parameter LMs; as such, relatively high task accuracies are expected. We use the label SENTIMENT for the in-distribution test set.

**SentimentOOD.** To evaluate the behavior of the estimators over out-of-distribution (OOD) data, we consider an additional evaluation set, SENTIMENTOOD, $|\mathcal{D}_{\text{te}}| = 4750$. We use the SemEval-2017 Task 4a test set (Rosenthal et al., 2017), which consists of short-form social media posts that differ in the distribution

of topics, language styles, and lengths relative to the movie reviews. We balance the test set, dropping the third class (neutral), setting the semantics of the true labels to be the same as that of the movie reviews: $y \in \{0 = \text{negative}, 1 = \text{positive}\}$.

**SentimentShuffled and SentimentOODShuffled.** In the Appendix, we consider two additional out-of-distribution challenge test sets, SENTIMENTSHUFFLED and SENTIMENTOODSHUFFLED, constructed by randomly shuffling the input documents for each of SENTIMENT and SENTIMENTOOD, respectively. The semantics of the original labels are maintained. This requires the models and estimators to attempt a sentiment classification over the bag-of-words input, or reject the classification. This represents the setting where an LM is given far out-of-distribution input, and additionally provides a control on test-set contamination of the underlying LMs, which due to the shuffling, are relatively unlikely to have seen all of the long, contiguous n-gram sequences from these documents in training or fine-tuning.

**Prompt.** For this task, we prompt the LMs for a binary classification (`Yes` or `No`) as follows: `Here is a movie review. <review> `DOCUMENT` </review> Is the sentiment of the movie review positive? Answer Yes if the sentiment is positive. Answer No if the sentiment is negative. Start your response with Yes or No.`

We replace `DOCUMENT` with the corresponding text for each instance.

## 4.2 Task: Factcheck

**Factcheck.** As a more challenging binary classification task for LMs, we consider the fact check data of Azaria & Mitchell (2023). The training and calibration sets, a combined total of 6k instances, consist of single sentence statements that have been semi-automatically generated via templates and a knowledge base. The task is to determine whether the statement is true or false, $y \in \{0 = \text{false}, 1 = \text{true}\}$. The held-out eval set (FACTCHECK), $|\mathcal{D}_{\text{te}}| = 245$, the focus of our analysis, has been constructed by having an LM generate a statement continued from a true statement not otherwise in the dataset. These evaluation statements are checked manually and assigned labels by human annotators. In addition to being a relatively challenging task that evaluates—at least in principle—the latent knowledge stored within an LM's parameters, the test set is representative of the types of co-variate shifts over high-dimensional inputs that can be problematic for real applications, and challenging to characterize without model assistance and ground-truth labels. It was observed in Azaria & Mitchell (2023) that the accuracy of existing LM classifiers is lower on this generated, held-out test set compared to the calibration set. However, these test sentences would seem to also be simple true-false statements, reflecting that it is not necessarily straightforward for a human user to detect distribution shifts over high-dimensional inputs. As such, we seek for our models and estimators to reflect such shifts via the predictive uncertainty.

**FactcheckShuffled.** As with the sentiment task, in the Appendix we also consider an additional out-of-distribution challenge test set, FACTCHECKSHUFFLED, constructed by randomly shuffling the input documents of FACTCHECK. Since the task prompt (below) seeks to classify errors, we set the ground-truth labels of the shuffled counterparts to $y = 0$.

**Prompt.** Similar to the sentiment task, we prompt the LMs for a binary classification (`Yes` or `No`) as follows: `Here is a statement that may contain errors. <statement> `DOCUMENT` </statement> Is the statement true? Answer Yes if the statement is true. Answer No if the statement is false. Start your response with Yes or No.`

We replace `DOCUMENT` with the corresponding text for each instance.

## 4.3 Models

We consider two representative, publicly-available decoder-only Transformer-based language models: The 3.8 billion-parameter `Phi-3.5-mini-instruct` model (PHI3.5) (Abdin et al., 2024), and the 47 billion-

parameter `Mixtral 8x7B Instruct v0.1` mixture-of-experts model (MIXTRAL8X7B) (Jiang et al., 2024). The parameters of these models stay fixed in all experiments.

**Hidden states.** For both models, we take as $\boldsymbol{h}$ the concatenation of the final-layer hidden state of the final sequence position (i.e., the hidden state that is the input to the linear-layer over the output vocabulary for the `Yes` or `No` generation) with the mean over all final-layer hidden states. For PHI3.5, this results in $\boldsymbol{h} \in \mathbb{R}^{6144}$, and for MIXTRAL8X7B, $\boldsymbol{h} \in \mathbb{R}^{8192}$.

### 4.4 Estimators

Holding each underlying LM constant, we examine representative classes of estimators used with neural networks, setting $\alpha = 0.95$ for all experiments. At the most basic, but also perhaps the most commonly used in practice, representing the absence of a post-hoc calibration method, we simply threshold the output, $\text{softmax}(\boldsymbol{z}) \geq \alpha$, where the temperature $\tau = 1$. For this, we use the label softmax. As an established empirical approach for calibrating neural networks, we provide a comparison to temperature scaling (Guo et al., 2017), a single parameter version of post-hoc Platt-scaling (Platt, 1999), with the label TEMPSCALING. In this case, the estimator is the thresholding of the output $\text{softmax}(\boldsymbol{z}; \tau) \geq \alpha$ after learning a value for $\tau$ over $\mathcal{D}_{\text{ca}}$. We also provide a comparison to two representative conformal predictors, the APS method of Romano et al. (2020) and the adaptiveness-optimized RAPS algorithm of Angelopoulos et al. (2021). The admission criteria for the APS and RAPS estimators is prediction sets of size 1, at the 0.05 level (i.e., $1 - \alpha$, as defined here). We consider these estimators over the logits corresponding to the `Yes` and `No` indexes of the output linear-layer of the underlying LM (PHI3.5 and MIXTRAL8X7B), which provides a reference point without introducing additional adaptor layers. We also consider these baselines over 1-D CNN adaptors over $\boldsymbol{h}$ of each LM (PHI3.5+ADAPTOR and MIXTRAL8X7B+ADAPTOR). These baseline adaptors are identical to those used in the corresponding SDM activation layers, with $M = 1000$, and are similarly trained for $J = 10$ iterations of 200 epochs, but unlike the SDM activation layers, the stopping criteria is the minimum balanced (across classes) average cross-entropy loss. In Appendix A.4, we further compare to variational Bayesian last-layer neural networks (VBLL), a computationally efficient Bayesian approach (Harrison et al., 2024).

We then compare to the final-layer SDM activations over $\boldsymbol{h}$ of each LM (PHI3.5+SDM and MIXTRAL8X7B+SDM). For reference, we provide the result of thresholding a softmax over these adaptors at $\alpha$, as above, as well as a thresholded softmax that simply treats $d$ as the inverse-temperature, $\text{SOFTMAX}(d \cdot \boldsymbol{z}')$, which is equivalent to setting $q = e - 2$ in the SDM activation. We also consider an analogous threshold over the activation output, $\text{SDM}(\boldsymbol{z}') \geq \alpha$, for which we use SDM as the estimator label. Finally, we use the label $\text{SDM}_{\text{HR}}$ for the selection criteria proposed in Eq. 9.

As a common point of reference, the label NO-REJECT refers to predictions without any selective filtering (i.e., the raw output accuracies, derived from the $\arg\max$ over the final linear-layer).

## 5 Results

**In-distribution data.** The results for the in-distribution SENTIMENT dataset appear in Table 1. Even on this in-distribution dataset, the estimators over the underlying LMs without adaptor layers exhibit modest over-confidence, which is reflected in conditional accuracies that fall below the expected $\alpha$. The estimators over the adaptor layers and the SDM activation all obtain the desired conditional accuracies, with the class-wise accuracies of the models themselves $\geq \alpha$, with differences arising in the proportion of admitted points. Here and elsewhere, $\text{SOFTMAX}(d \cdot \boldsymbol{z}')$ tends to be overly conservative in rejecting points. To be expected, the $\text{SDM}_{\text{HR}}$ estimator tends to be more conservative than simply thresholding the SDM activation on the in-distribution data, but the latter lacks the assurances on the class-conditional accuracy obtained by the constraints on the HIGH-RELIABILITY region. In practice, this behavior can be used as a basis to triage selective classifications: For example, documents in the HIGH-RELIABILITY region might be treated as automated, or semi-automated, predictions in the decision pipeline, whereas other documents might be triaged by $\text{SDM}(\boldsymbol{z}')_{\hat{y}}$ for calling more resource-intensive LM tools, or human adjudication. Importantly, as we discuss below with the co-variate-shifted and out-of-distribution datasets, just using the softmax or the other estimators does not provide a reliable substrate for basing such conditional branching decisions.

Table 1: Comparison of estimators for the sentiment datasets, with $\alpha = 0.95$. R indicates all predictions were rejected, which is preferred over falling under the expected accuracy. $n = |\text{Admitted}|$, the count of non-rejected documents.

| | | | Class-conditional | | | | Prediction-conditional | | | | Marginal | |
| | | | $y = 0$ | | $y = 1$ | | $\hat{y} = 0$ | | $\hat{y} = 1$ | | $y \in \{0,1\}$ | |
| Dataset | Model | Estimator | Acc. | $\frac{n}{|\mathcal{D}_{te}|}$ | Acc. | $\frac{n}{|\mathcal{D}_{te}|}$ | Acc. | $\frac{n}{|\mathcal{D}_{te}|}$ | Acc. | $\frac{n}{|\mathcal{D}_{te}|}$ | Acc. | $\frac{n}{|\mathcal{D}_{te}|}$ |
|---|---|---|---|---|---|---|---|---|---|---|---|---|
| SENTIMENT | PHI3.5 | NO-REJECT | 0.98 | 0.50 | 0.85 | 0.50 | 0.86 | 0.57 | 0.98 | 0.43 | 0.91 | 1. |
| SENTIMENT | PHI3.5 | SOFTMAX | 0.98 | 0.50 | 0.86 | 0.48 | 0.88 | 0.56 | 0.98 | 0.42 | 0.93 | 0.98 |
| SENTIMENT | PHI3.5 | TEMPSCALING | 0.99 | 0.49 | 0.91 | 0.41 | 0.93 | 0.52 | 0.99 | 0.38 | 0.95 | 0.90 |
| SENTIMENT | PHI3.5 | APS | 0.99 | 0.49 | 0.92 | 0.40 | 0.94 | 0.51 | 0.99 | 0.37 | 0.96 | 0.89 |
| SENTIMENT | PHI3.5 | RAPS | 0.99 | 0.48 | 0.91 | 0.41 | 0.93 | 0.51 | 0.99 | 0.38 | 0.95 | 0.90 |
| SENTIMENT | PHI3.5+ADAPTOR | NO-REJECT | 0.97 | 0.50 | 0.95 | 0.50 | 0.96 | 0.51 | 0.97 | 0.49 | 0.96 | 1. |
| SENTIMENT | PHI3.5+ADAPTOR | SOFTMAX | 0.99 | 0.42 | 1.00 | 0.42 | 1.00 | 0.42 | 0.99 | 0.42 | 0.99 | 0.84 |
| SENTIMENT | PHI3.5+ADAPTOR | TEMPSCALING | 0.99 | 0.42 | 1.00 | 0.41 | 1.00 | 0.42 | 0.99 | 0.41 | 0.99 | 0.83 |
| SENTIMENT | PHI3.5+ADAPTOR | APS | 0.98 | 0.45 | 0.98 | 0.45 | 0.98 | 0.45 | 0.98 | 0.45 | 0.98 | 0.90 |
| SENTIMENT | PHI3.5+ADAPTOR | RAPS | 0.98 | 0.45 | 0.98 | 0.44 | 0.98 | 0.45 | 0.98 | 0.44 | 0.98 | 0.89 |
| SENTIMENT | PHI3.5+SDM | NO-REJECT | 0.96 | 0.50 | 0.96 | 0.50 | 0.96 | 0.50 | 0.96 | 0.50 | 0.96 | 1. |
| SENTIMENT | PHI3.5+SDM | SOFTMAX | 0.97 | 0.48 | 0.97 | 0.48 | 0.97 | 0.48 | 0.97 | 0.48 | 0.97 | 0.96 |
| SENTIMENT | PHI3.5+SDM | SOFTMAX($d \cdot z'$) | 0.99 | 0.30 | 0.99 | 0.24 | 1.00 | 0.30 | 0.99 | 0.24 | 0.99 | 0.54 |
| SENTIMENT | PHI3.5+SDM | SDM | 0.99 | 0.43 | 0.99 | 0.38 | 0.99 | 0.43 | 0.99 | 0.38 | 0.99 | 0.81 |
| SENTIMENT | PHI3.5+SDM | SDM$_{HR}$ | 1.00 | 0.37 | 0.99 | 0.30 | 0.99 | 0.38 | 1.00 | 0.30 | 0.99 | 0.68 |
| SENTIMENT | MIXTRAL8X7B | NO-REJECT | 0.98 | 0.50 | 0.88 | 0.50 | 0.89 | 0.55 | 0.98 | 0.45 | 0.93 | 1. |
| SENTIMENT | MIXTRAL8X7B | SOFTMAX | 0.98 | 0.50 | 0.88 | 0.50 | 0.89 | 0.55 | 0.98 | 0.45 | 0.93 | 1.00 |
| SENTIMENT | MIXTRAL8X7B | TEMPSCALING | 0.99 | 0.50 | 0.90 | 0.48 | 0.91 | 0.54 | 0.98 | 0.44 | 0.94 | 0.98 |
| SENTIMENT | MIXTRAL8X7B | APS | 0.98 | 0.49 | 0.91 | 0.47 | 0.92 | 0.52 | 0.98 | 0.44 | 0.95 | 0.96 |
| SENTIMENT | MIXTRAL8X7B | RAPS | 0.99 | 0.49 | 0.92 | 0.47 | 0.93 | 0.52 | 0.98 | 0.44 | 0.95 | 0.96 |
| SENTIMENT | MIXTRAL8X7B+ADAPTOR | NO-REJECT | 0.97 | 0.50 | 0.96 | 0.50 | 0.96 | 0.51 | 0.97 | 0.49 | 0.97 | 1. |
| SENTIMENT | MIXTRAL8X7B+ADAPTOR | SOFTMAX | 0.99 | 0.45 | 0.99 | 0.43 | 0.99 | 0.45 | 0.99 | 0.43 | 0.99 | 0.87 |
| SENTIMENT | MIXTRAL8X7B+ADAPTOR | TEMPSCALING | 0.99 | 0.43 | 0.99 | 0.41 | 0.99 | 0.43 | 0.99 | 0.41 | 0.99 | 0.84 |
| SENTIMENT | MIXTRAL8X7B+ADAPTOR | APS | 0.99 | 0.46 | 0.98 | 0.45 | 0.98 | 0.46 | 0.99 | 0.44 | 0.99 | 0.91 |
| SENTIMENT | MIXTRAL8X7B+ADAPTOR | RAPS | 0.99 | 0.46 | 0.98 | 0.45 | 0.98 | 0.47 | 0.98 | 0.45 | 0.98 | 0.92 |
| SENTIMENT | MIXTRAL8X7B+SDM | NO-REJECT | 0.96 | 0.50 | 0.95 | 0.50 | 0.95 | 0.51 | 0.96 | 0.49 | 0.96 | 1. |
| SENTIMENT | MIXTRAL8X7B+SDM | SOFTMAX | 0.97 | 0.49 | 0.96 | 0.49 | 0.96 | 0.50 | 0.97 | 0.49 | 0.97 | 0.98 |
| SENTIMENT | MIXTRAL8X7B+SDM | SOFTMAX($d \cdot z'$) | 0.99 | 0.43 | 0.99 | 0.33 | 0.99 | 0.43 | 0.99 | 0.33 | 0.99 | 0.77 |
| SENTIMENT | MIXTRAL8X7B+SDM | SDM | 0.98 | 0.48 | 0.98 | 0.43 | 0.98 | 0.47 | 0.98 | 0.43 | 0.98 | 0.90 |
| SENTIMENT | MIXTRAL8X7B+SDM | SDM$_{HR}$ | 0.99 | 0.41 | 0.98 | 0.33 | 0.99 | 0.41 | 0.98 | 0.33 | 0.99 | 0.74 |
| SENTIMENTOOD | PHI3.5 | NO-REJECT | 1.00 | 0.50 | 0.53 | 0.50 | 0.68 | 0.73 | 0.99 | 0.27 | 0.76 | 1. |
| SENTIMENTOOD | PHI3.5 | SOFTMAX | 1.00 | 0.50 | 0.54 | 0.46 | 0.70 | 0.71 | 0.99 | 0.25 | 0.78 | 0.96 |
| SENTIMENTOOD | PHI3.5 | TEMPSCALING | 1.00 | 0.49 | 0.58 | 0.30 | 0.80 | 0.62 | 0.99 | 0.17 | 0.84 | 0.79 |
| SENTIMENTOOD | PHI3.5 | APS | 1.00 | 0.49 | 0.59 | 0.28 | 0.81 | 0.60 | 0.99 | 0.17 | 0.85 | 0.77 |
| SENTIMENTOOD | PHI3.5 | RAPS | 1.00 | 0.49 | 0.59 | 0.28 | 0.81 | 0.60 | 0.99 | 0.17 | 0.85 | 0.77 |
| SENTIMENTOOD | PHI3.5+ADAPTOR | NO-REJECT | 0.47 | 0.50 | 0.70 | 0.50 | 0.61 | 0.38 | 0.57 | 0.62 | 0.59 | 1. |
| SENTIMENTOOD | PHI3.5+ADAPTOR | SOFTMAX | 0.57 | 0.03 | 0.96 | 0.07 | 0.84 | 0.02 | 0.85 | 0.07 | 0.85 | 0.09 |
| SENTIMENTOOD | PHI3.5+ADAPTOR | TEMPSCALING | 0.60 | 0.02 | 0.97 | 0.05 | 0.86 | 0.01 | 0.87 | 0.06 | 0.87 | 0.07 |
| SENTIMENTOOD | PHI3.5+ADAPTOR | APS | 0.46 | 0.14 | 0.83 | 0.18 | 0.67 | 0.09 | 0.68 | 0.22 | 0.68 | 0.32 |
| SENTIMENTOOD | PHI3.5+ADAPTOR | RAPS | 0.48 | 0.13 | 0.82 | 0.18 | 0.66 | 0.10 | 0.68 | 0.22 | 0.68 | 0.32 |
| SENTIMENTOOD | PHI3.5+SDM | NO-REJECT | 0.92 | 0.50 | 0.84 | 0.50 | 0.85 | 0.54 | 0.91 | 0.46 | 0.88 | 1. |
| SENTIMENTOOD | PHI3.5+SDM | SOFTMAX | 0.96 | 0.42 | 0.87 | 0.45 | 0.87 | 0.46 | 0.96 | 0.41 | 0.91 | 0.87 |
| SENTIMENTOOD | PHI3.5+SDM | SOFTMAX($d \cdot z'$) | 1. | <0.01 | 1. | <0.01 | 1. | <0.01 | 1. | <0.01 | 1. | <0.01 |
| SENTIMENTOOD | PHI3.5+SDM | SDM | 1. | 0.01 | 0.98 | 0.01 | 0.98 | 0.01 | 1. | 0.01 | 0.99 | 0.02 |
| SENTIMENTOOD | PHI3.5+SDM | SDM$_{HR}$ | 1. | <0.01 | 1. | <0.01 | 1. | <0.01 | 1. | <0.01 | 1. | 0.01 |
| SENTIMENTOOD | MIXTRAL8X7B | NO-REJECT | 1.00 | 0.50 | 0.35 | 0.50 | 0.61 | 0.82 | 1.00 | 0.18 | 0.67 | 1. |
| SENTIMENTOOD | MIXTRAL8X7B | SOFTMAX | 1.00 | 0.50 | 0.35 | 0.49 | 0.61 | 0.82 | 1.00 | 0.17 | 0.68 | 0.99 |
| SENTIMENTOOD | MIXTRAL8X7B | TEMPSCALING | 1.00 | 0.49 | 0.37 | 0.41 | 0.66 | 0.75 | 0.99 | 0.15 | 0.71 | 0.90 |
| SENTIMENTOOD | MIXTRAL8X7B | APS | 1.00 | 0.45 | 0.44 | 0.32 | 0.71 | 0.63 | 0.99 | 0.14 | 0.77 | 0.77 |
| SENTIMENTOOD | MIXTRAL8X7B | RAPS | 1.00 | 0.45 | 0.44 | 0.32 | 0.72 | 0.63 | 0.99 | 0.14 | 0.77 | 0.77 |
| SENTIMENTOOD | MIXTRAL8X7B+ADAPTOR | NO-REJECT | 0.88 | 0.50 | 0.51 | 0.50 | 0.64 | 0.69 | 0.82 | 0.31 | 0.70 | 1. |
| SENTIMENTOOD | MIXTRAL8X7B+ADAPTOR | SOFTMAX | 0.98 | 0.02 | 0.83 | 0.07 | 0.66 | 0.04 | 0.99 | 0.06 | 0.87 | 0.10 |
| SENTIMENTOOD | MIXTRAL8X7B+ADAPTOR | TEMPSCALING | 0.98 | 0.01 | 0.90 | 0.05 | 0.67 | 0.02 | 1.00 | 0.05 | 0.91 | 0.06 |
| SENTIMENTOOD | MIXTRAL8X7B+ADAPTOR | APS | 0.94 | 0.14 | 0.63 | 0.18 | 0.67 | 0.20 | 0.93 | 0.12 | 0.77 | 0.32 |
| SENTIMENTOOD | MIXTRAL8X7B+ADAPTOR | RAPS | 0.94 | 0.14 | 0.63 | 0.18 | 0.67 | 0.20 | 0.93 | 0.12 | 0.76 | 0.32 |
| SENTIMENTOOD | MIXTRAL8X7B+SDM | NO-REJECT | 0.71 | 0.50 | 0.83 | 0.50 | 0.81 | 0.44 | 0.74 | 0.56 | 0.77 | 1. |
| SENTIMENTOOD | MIXTRAL8X7B+SDM | SOFTMAX | 0.74 | 0.43 | 0.86 | 0.47 | 0.83 | 0.39 | 0.78 | 0.52 | 0.80 | 0.91 |
| SENTIMENTOOD | MIXTRAL8X7B+SDM | SOFTMAX($d \cdot z'$) | 1. | <0.01 | 0.98 | 0.02 | 0.78 | <0.01 | 1. | 0.02 | 0.98 | 0.02 |
| SENTIMENTOOD | MIXTRAL8X7B+SDM | SDM | 0.98 | 0.05 | 0.96 | 0.04 | 0.97 | 0.05 | 0.98 | 0.04 | 0.97 | 0.08 |
| SENTIMENTOOD | MIXTRAL8X7B+SDM | SDM$_{HR}$ | 0.9487 | 0.01 | 0.96 | 0.01 | 0.9487 | 0.01 | 0.96 | 0.01 | 0.95 | 0.02 |

Table 2: Comparison of estimators for the factcheck datasets. Unless specified otherwise, $\alpha = 0.95$. R indicates all predictions were rejected, which is preferred over falling under the expected accuracy. $n = |\text{Admitted}|$, the count of non-rejected documents.

| | | | Class-conditional | | | | Prediction-conditional | | | | Marginal | |
| | | | $y = 0$ | | $y = 1$ | | $\hat{y} = 0$ | | $\hat{y} = 1$ | | $y \in \{0,1\}$ | |
| Dataset | Model | Estimator | Acc. | $\frac{n}{|\mathcal{D}_{\text{te}}|}$ | Acc. | $\frac{n}{|\mathcal{D}_{\text{te}}|}$ | Acc. | $\frac{n}{|\mathcal{D}_{\text{te}}|}$ | Acc. | $\frac{n}{|\mathcal{D}_{\text{te}}|}$ | Acc. | $\frac{n}{|\mathcal{D}_{\text{te}}|}$ |
|---|---|---|---|---|---|---|---|---|---|---|---|---|
| FACTCHECK | PHI3.5 | NO-REJECT | 0.94 | 0.51 | 0.71 | 0.49 | 0.78 | 0.62 | 0.92 | 0.38 | 0.83 | 1. |
| FACTCHECK | PHI3.5 | SOFTMAX | 0.94 | 0.51 | 0.73 | 0.46 | 0.79 | 0.60 | 0.92 | 0.36 | 0.84 | 0.97 |
| FACTCHECK | PHI3.5 | TEMPSCALING | 0.97 | 0.38 | 0.79 | 0.37 | 0.83 | 0.45 | 0.96 | 0.31 | 0.88 | 0.76 |
| FACTCHECK | PHI3.5 | APS | 0.98 | 0.22 | 0.82 | 0.27 | 0.82 | 0.27 | 0.98 | 0.23 | 0.89 | 0.50 |
| FACTCHECK | PHI3.5 | RAPS | 0.98 | 0.20 | 0.84 | 0.28 | 0.81 | 0.24 | 0.98 | 0.24 | 0.90 | 0.47 |
| FACTCHECK | PHI3.5+ADAPTOR | NO-REJECT | 0.33 | 0.51 | 0.94 | 0.49 | 0.85 | 0.20 | 0.57 | 0.80 | 0.62 | 1. |
| FACTCHECK | PHI3.5+ADAPTOR | SOFTMAX | 0.40 | 0.08 | 0.99 | 0.33 | 0.89 | 0.04 | 0.87 | 0.37 | 0.87 | 0.41 |
| FACTCHECK | PHI3.5+ADAPTOR | TEMPSCALING | 0.38 | 0.07 | 0.99 | 0.29 | 0.86 | 0.03 | 0.88 | 0.33 | 0.88 | 0.36 |
| FACTCHECK | PHI3.5+ADAPTOR | APS | 0.26 | 0.14 | 0.99 | 0.38 | 0.90 | 0.04 | 0.78 | 0.48 | 0.79 | 0.52 |
| FACTCHECK | PHI3.5+ADAPTOR | RAPS | 0.36 | 0.18 | 0.98 | 0.35 | 0.89 | 0.07 | 0.74 | 0.46 | 0.76 | 0.53 |
| FACTCHECK | PHI3.5+SDM | NO-REJECT | 0.70 | 0.51 | 0.88 | 0.49 | 0.86 | 0.42 | 0.73 | 0.58 | 0.79 | 1. |
| FACTCHECK | PHI3.5+SDM | SOFTMAX | 0.75 | 0.27 | 0.94 | 0.39 | 0.89 | 0.22 | 0. 85 | 0.43 | 0.86 | 0.65 |
| FACTCHECK | PHI3.5+SDM | SOFTMAX($d \cdot \boldsymbol{z}'$) | R | 0. | 1. | 0.03 | R | 0. | 1. | 0.03 | 1. | 0.03 |
| FACTCHECK | PHI3.5+SDM | SDM | 1. | 0.01 | 0.97 | 0.14 | 0.75 | 0.02 | 1. | 0.14 | 0.97 | 0.16 |
| FACTCHECK | PHI3.5+SDM | SDM$_{\text{HR}}$ | R | 0. | 1. | 0.12 | R | 0. | 1. | 0.12 | 1. | 0.12 |
| FACTCHECK | MIXTRAL8x7B | NO-REJECT | 0.98 | 0.51 | 0.48 | 0.49 | 0.66 | 0.76 | 0.95 | 0.24 | 0.73 | 1. |
| FACTCHECK | MIXTRAL8x7B | SOFTMAX | 0.98 | 0.51 | 0.48 | 0.49 | 0.66 | 0.76 | 0.95 | 0.24 | 0.73 | 1. |
| FACTCHECK | MIXTRAL8x7B | TEMPSCALING | 0.99 | 0.50 | 0.46 | 0.43 | 0.68 | 0.73 | 0.98 | 0.20 | 0.75 | 0.93 |
| FACTCHECK | MIXTRAL8x7B | APS | 1. | 0.18 | 0.80 | 0.16 | 0.84 | 0.21 | 1. | 0.13 | 0.90 | 0.34 |
| FACTCHECK | MIXTRAL8x7B | RAPS | 1. | 0.14 | 0.66 | 0.20 | 0.67 | 0.21 | 1. | 0.13 | 0.80 | 0.35 |
| FACTCHECK | MIXTRAL8x7B+ADAPTOR | NO-REJECT | 0.56 | 0.51 | 0.87 | 0.49 | 0.82 | 0.36 | 0.65 | 0.64 | 0.71 | 1. |
| FACTCHECK | MIXTRAL8x7B+ADAPTOR | SOFTMAX | 0.68 | 0.11 | 0.97 | 0.31 | 0.90 | 0.09 | 0.89 | 0.34 | 0.89 | 0.42 |
| FACTCHECK | MIXTRAL8x7B+ADAPTOR | TEMPSCALING | 0.70 | 0.09 | 0.97 | 0.29 | 0.89 | 0.07 | 0.91 | 0.31 | 0.91 | 0.39 |
| FACTCHECK | MIXTRAL8x7B+ADAPTOR | APS | 0.62 | 0.22 | 0.96 | 0.37 | 0.89 | 0.16 | 0.80 | 0.44 | 0.83 | 0.59 |
| FACTCHECK | MIXTRAL8x7B+ADAPTOR | RAPS | 0.65 | 0.22 | 0.96 | 0.35 | 0.92 | 0.16 | 0.81 | 0.41 | 0.84 | 0.57 |
| FACTCHECK | MIXTRAL8x7B+SDM | NO-REJECT | 0.63 | 0.51 | 0.90 | 0.49 | 0.87 | 0.38 | 0.70 | 0.62 | 0.76 | 1. |
| FACTCHECK | MIXTRAL8x7B+SDM | SOFTMAX | 0.67 | 0.34 | 0.96 | 0.40 | 0.93 | 0.24 | 0.78 | 0.50 | 0.83 | 0.74 |
| FACTCHECK | MIXTRAL8x7B+SDM | SOFTMAX($d \cdot \boldsymbol{z}'$) | R | 0. | 0.80 | 0.04 | 0. | 0.01 | 1. | 0.03 | 0.80 | 0.04 |
| FACTCHECK | MIXTRAL8x7B+SDM | SDM | 0.88 | 0.10 | 0.95 | 0.18 | 0.91 | 0.09 | 0.93 | 0.18 | 0.93 | 0.27 |
| FACTCHECK | MIXTRAL8x7B+SDM | SDM$_{\text{HR}}$ | R | 0. | R | 0. | R | 0. | R | 0. | R | 0. |
| FACTCHECK | MIXTRAL8x7B+SDM | SDM, $\alpha = 0.94$ | 0.85 | 0.11 | 0.95 | 0.18 | 0.92 | 0.10 | 0.91 | 0.19 | 0.91 | 0.29 |
| FACTCHECK | MIXTRAL8x7B+SDM | SDM$_{\text{HR}}$, $\alpha = 0.94$ | 1. | 0.03 | 0.95 | 0.16 | 0.80 | 0.04 | 1. | 0.15 | 0.96 | 0.19 |

Table 3: Reference results over $\mathcal{D}_{\mathrm{ca}}$ to illustrate the behavior of $q'_{\min}$. The value of $q'_{\min}$ tends to increase as the accuracy over $\mathcal{D}_{\mathrm{ca}}$ decreases, reflecting a more conservative HIGH-RELIABILITY region. Alg. 1 failed to find a finite $q'_{\min}$ for MIXTRAL8X7B+SDM over the FACTCHECK calibration set at $\alpha = 0.95$, so for reference, we also show the HIGH-RELIABILITY region at $\alpha = 0.94$.

| | | | Class-conditional | | | | Prediction-conditional | | | | Marginal | |
| | | | $y = 0$ | | $y = 1$ | | $\hat{y} = 0$ | | $\hat{y} = 1$ | | $y \in \{0,1\}$ | |
| Dataset | Model | Estimator | ACC. | $\frac{n}{\|\mathcal{D}_{\mathrm{te}}\|}$ | ACC. | $\frac{n}{\|\mathcal{D}_{\mathrm{te}}\|}$ | ACC. | $\frac{n}{\|\mathcal{D}_{\mathrm{te}}\|}$ | ACC. | $\frac{n}{\|\mathcal{D}_{\mathrm{te}}\|}$ | ACC. | $\frac{n}{\|\mathcal{D}_{\mathrm{te}}\|}$ |
|---|---|---|---|---|---|---|---|---|---|---|---|---|
| SENTIMENT $\mathcal{D}_{\mathrm{ca}}$ | PHI3.5+SDM | NO-REJECT | 0.95 | 0.50 | 0.96 | 0.50 | 0.96 | 0.50 | 0.95 | 0.50 | 0.96 | 1. |
| SENTIMENT $\mathcal{D}_{\mathrm{ca}}$ | PHI3.5+SDM | SOFTMAX | 0.96 | 0.48 | 0.97 | 0.48 | 0.97 | 0.47 | 0.96 | 0.49 | 0.97 | 0.96 |
| SENTIMENT $\mathcal{D}_{\mathrm{ca}}$ | PHI3.5+SDM | SOFTMAX$(d \cdot \boldsymbol{z}')$ | 0.99 | 0.31 | 0.99 | 0.24 | 1.00 | 0.31 | 0.99 | 0.24 | 0.99 | 0.55 |
| SENTIMENT $\mathcal{D}_{\mathrm{ca}}$ | PHI3.5+SDM | SDM | 0.99 | 0.42 | 0.99 | 0.39 | 0.99 | 0.42 | 0.99 | 0.39 | 0.99 | 0.81 |
| SENTIMENT $\mathcal{D}_{\mathrm{ca}}$ | PHI3.5+SDM | SDM$_{\mathrm{HR}}, \alpha = 0.95, q'_{\min} = 52.2$ | 0.99 | 0.38 | 0.99 | 0.32 | 0.99 | 0.38 | 0.99 | 0.31 | 0.99 | 0.69 |
| SENTIMENT $\mathcal{D}_{\mathrm{ca}}$ | MIXTRAL8X7B+SDM | NO-REJECT | 0.96 | 0.50 | 0.96 | 0.50 | 0.96 | 0.50 | 0.96 | 0.50 | 0.96 | 1. |
| SENTIMENT $\mathcal{D}_{\mathrm{ca}}$ | MIXTRAL8X7B+SDM | SOFTMAX | 0.96 | 0.49 | 0.97 | 0.49 | 0.97 | 0.49 | 0.96 | 0.49 | 0.97 | 0.98 |
| SENTIMENT $\mathcal{D}_{\mathrm{ca}}$ | MIXTRAL8X7B+SDM | SOFTMAX$(d \cdot \boldsymbol{z}')$ | 1.00 | 0.43 | 0.99 | 0.35 | 0.99 | 0.43 | 1.00 | 0.34 | 0.99 | 0.78 |
| SENTIMENT $\mathcal{D}_{\mathrm{ca}}$ | MIXTRAL8X7B+SDM | SDM | 0.99 | 0.47 | 0.98 | 0.43 | 0.98 | 0.47 | 0.98 | 0.43 | 0.98 | 0.90 |
| SENTIMENT $\mathcal{D}_{\mathrm{ca}}$ | MIXTRAL8X7B+SDM | SDM$_{\mathrm{HR}}, \alpha = 0.95, q'_{\min} = 63.0$ | 0.99 | 0.41 | 0.99 | 0.34 | 0.99 | 0.41 | 0.99 | 0.34 | 0.99 | 0.74 |
| FACTCHECK $\mathcal{D}_{\mathrm{ca}}$ | PHI3.5+SDM | NO-REJECT | 0.90 | 0.50 | 0.91 | 0.50 | 0.91 | 0.49 | 0.90 | 0.51 | 0.90 | 1. |
| FACTCHECK $\mathcal{D}_{\mathrm{ca}}$ | PHI3.5+SDM | SOFTMAX | 0.94 | 0.32 | 0.96 | 0.41 | 0.95 | 0.31 | 0.96 | 0.41 | 0.96 | 0.72 |
| FACTCHECK $\mathcal{D}_{\mathrm{ca}}$ | PHI3.5+SDM | SOFTMAX$(d \cdot \boldsymbol{z}')$ | 0.98 | 0.08 | 1.00 | 0.07 | 1.00 | 0.08 | 0.98 | 0.07 | 0.99 | 0.15 |
| FACTCHECK $\mathcal{D}_{\mathrm{ca}}$ | PHI3.5+SDM | SDM | 0.98 | 0.33 | 0.99 | 0.27 | 0.99 | 0.33 | 0.98 | 0.27 | 0.98 | 0.60 |
| FACTCHECK $\mathcal{D}_{\mathrm{ca}}$ | PHI3.5+SDM | SDM$_{\mathrm{HR}}, \alpha = 0.95, q'_{\min} = 95.0$ | 1.00 | 0.19 | 0.99 | 0.12 | 1.00 | 0.19 | 0.99 | 0.12 | 1.00 | 0.31 |
| FACTCHECK $\mathcal{D}_{\mathrm{ca}}$ | MIXTRAL8X7B+SDM | NO-REJECT | 0.90 | 0.50 | 0.92 | 0.50 | 0.92 | 0.49 | 0.90 | 0.51 | 0.91 | 1. |
| FACTCHECK $\mathcal{D}_{\mathrm{ca}}$ | MIXTRAL8X7B+SDM | SOFTMAX | 0.94 | 0.44 | 0.96 | 0.45 | 0.96 | 0.43 | 0.94 | 0.46 | 0.95 | 0.89 |
| FACTCHECK $\mathcal{D}_{\mathrm{ca}}$ | MIXTRAL8X7B+SDM | SOFTMAX$(d \cdot \boldsymbol{z}')$ | 0.98 | 0.28 | 0.98 | 0.17 | 0.99 | 0.27 | 0.96 | 0.17 | 0.98 | 0.44 |
| FACTCHECK $\mathcal{D}_{\mathrm{ca}}$ | MIXTRAL8X7B+SDM | SDM | 0.97 | 0.41 | 0.95 | 0.32 | 0.96 | 0.41 | 0.95 | 0.32 | 0.96 | 0.73 |
| FACTCHECK $\mathcal{D}_{\mathrm{ca}}$ | MIXTRAL8X7B+SDM | SDM$_{\mathrm{HR}}, \alpha = 0.95, q'_{\min} = \infty$ | R | 0. | R | 0. | R | 0. | R | 0. | R | 0. |
| FACTCHECK $\mathcal{D}_{\mathrm{ca}}$ | MIXTRAL8X7B+SDM | SDM$, \alpha = 0.94$ | 0.96 | 0.42 | 0.95 | 0.33 | 0.96 | 0.42 | 0.95 | 0.32 | 0.96 | 0.74 |
| FACTCHECK $\mathcal{D}_{\mathrm{ca}}$ | MIXTRAL8X7B+SDM | SDM$_{\mathrm{HR}}, \alpha = 0.94, q'_{\min} = 134.0$ | 1.00 | 0.33 | 0.96 | 0.14 | 0.99 | 0.33 | 0.99 | 0.13 | 0.99 | 0.46 |

**Co-variate-shifted and Out-of-distribution data.** The bottom half of Table 1 provides the results for SENTIMENTOOD. Here, the distinctions between the estimators become clear, with the non-SDM-based estimators performing poorly, even in terms of the marginal accuracy. That would come as a surprise to end-users, whereas with the SDM and SDM$_{\mathrm{HR}}$ estimators, the out-of-distribution documents are more reliably rejected, with the few admitted predictions generally obtaining high conditional accuracies, despite the relatively low accuracies of the test set without selection (see NO-REJECT). A similar pattern is observed over the FACTCHECK dataset in Table 2.

**Understanding $q'_{\min}$.** For reference, Table 3 provides the results over $\mathcal{D}_{\mathrm{ca}}$ for the SDM-based estimators. The value of $q'_{\min}$ tends to increase as the accuracy over $\mathcal{D}_{\mathrm{ca}}$ decreases, reflecting a more conservative HIGH-RELIABILITY region that admits fewer points. Alg. 1 failed to find a finite $q'_{\min}$ for MIXTRAL8X7B+SDM over the FACTCHECK calibration set at $\alpha = 0.95$, so for reference, we also show the HIGH-RELIABILITY region at $\alpha = 0.94$, as well as in Table 2. In this way, $q'_{\min}$ provides a principled, data-driven indicator of the reliability of the estimates, which is interpretable as a simple indicator as to whether the conditional accuracies are, or are not, obtainable over $\mathcal{D}_{\mathrm{ca}}$ at the specified $\alpha$.

**Far out-of-distribution data.** The Appendix provides Table 4 and Table 5 with analogous results for the SENTIMENTSHUFFLED, SENTIMENTOODSHUFFLED, and FACTCHECKSHUFFLED datasets. The selection criteria of Eq. 9 reliably rejects the challenging predictions, whereas the non-SDM-based estimators fare poorly, in general. In this way, the SDM activation serves as an effective out-of-distribution detection method. With existing methods, defining an out-of-distribution point has been task- and problem-specific, and generally challenging over high-dimensional inputs, typically requiring additional modeling beyond that of the calibration or selection method. In contrast, Eq. 9 provides a principled approach for determining such cut-offs in a data- and model-driven manner, with minimal hyper-parameters, resulting in a separation of points over which the estimator tends to be reliable (namely, the admitted points) and those over which the estimates themselves tend to be unreliable.

## 6    Conclusion

We introduced SDM activation functions and SDM estimators, which are more robust estimators of the predictive uncertainty than those based on the commonly used softmax function. In this way, SDM activations provide a principled, data-driven substrate for approaching selective classification, calibration, and out-of-distribution detection with language models.

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

# A  Appendix

We provide the results for the far out-of-distribution (OOD) shuffled datasets in Appendix A.1. Additional implementation details are included in Appendix A.2. Appendix A.3 provides an approach for analyzing the effective sample size for both the class-conditional and prediction-conditional estimates. We close with Appendix A.4, which provides a comparison to the variational Bayesian last-layer estimators introduced in Harrison et al. (2024).

## A.1  Far OOD Shuffled Datasets

Table 4 shows results for the SENTIMENTSHUFFLED and SENTIMENTOODSHUFFLED datasets, and Table 5 shows results for the FACTCHECKSHUFFLED datasets, as discussed in Section 5.

## A.2  Additional Implementation Details

Replication code is available at the following URL: ANONYMIZED.

We mean center the input to $g$, the 1-D CNN of the SDM activation layer and the otherwise identical CNN adaptors of the baseline comparison estimators, via the mean and standard deviation over $\mathcal{D}_{\mathrm{tr}}$. In all experiments with adaptor layers, $M = 1000$ and we use a mini-batch size of 50. We use the Adam optimizer (Kingma & Ba, 2017) (without weight decay) with a learning rate of $1 \times 10^{-5}$ for training.

### A.2.1  Implementation of the SDM Activation Function

As is typical with implementations of the softmax function, for numerical stability, rather than directly calculating $\mathrm{SDM}(\boldsymbol{z}')$, we instead use the equivalent $\mathrm{SDM}(\boldsymbol{z}' - \max(\boldsymbol{z}'))$, shifting the input vector by its maximum value.

### A.2.2  Implementation of the Empirical CDF Function

The empirical CDF functions are assumed to be implemented such that the distance quantiles are exclusionary at the boundaries. When $d_{\mathrm{nearest}} = 0$, the $1 - \mathrm{eCDF}_{\mathrm{ca}}(d_{\mathrm{nearest}})$ quantile should be 1, and when $d_{\mathrm{nearest}}$ is greater than the maximum observed distance (across $\mathcal{D}_{\mathrm{ca}}$ for $\boldsymbol{x} \in \mathcal{D}_{\mathrm{te}}$ and $\boldsymbol{x} \in \mathcal{D}_{\mathrm{ca}}$, and across $\mathcal{D}_{\mathrm{tr}}$ for $\boldsymbol{x} \in \mathcal{D}_{\mathrm{tr}}$, the latter case only occurring during training), the $1 - \mathrm{eCDF}_{\mathrm{ca}}(d_{\mathrm{nearest}})$ quantile should be 0.

## A.3  Analyzing the Effective Sample Size

In the context of the SDM estimator, to parameterize the prior belief that data points with a looser connection to $\mathcal{D}_{\mathrm{tr}}$ reflect smaller effective sample sizes, while also explicitly accounting for the count of observed points in $\mathcal{D}_{\mathrm{ca}}$, the effective sample size for each test instance can be estimated with the following conservative assumption:

**Assumption A.1.** The effective sample size is increasing in $q'$, class-wise over $\mathcal{D}_{\mathrm{ca}}$.

For each $\boldsymbol{x} \in \mathcal{D}_{\mathrm{te}}$, using $q'$, we calculate $\hat{\mathbf{n}}$, the vector of effective sample sizes across classes, relative to $\mathcal{D}_{\mathrm{ca}}$, as:

$$\hat{\mathbf{n}} = [\,|\,\mathcal{D}_{\mathrm{ca}}\,|^{y_1} \cdot \mathrm{eCDF}_{\mathrm{ca}}^{y_1}(q'), \ldots, |\,\mathcal{D}_{\mathrm{ca}}\,|^{y_C} \cdot \mathrm{eCDF}_{\mathrm{ca}}^{y_C}(q')] \tag{10}$$

where $|\,\mathcal{D}_{\mathrm{ca}}\,|^{y_c}$ is the count of calibration set points with true label $y = c$.

The estimate of the effective sample size for each label can then be used to estimate the Beta-distributed error term of split-conformal coverage (Vovk, 2012), providing a sample-size-based error estimate for the class-conditional estimate, assuming exchangeability.

For the prediction-conditional estimates, assuming independent and identically distributed (i.i.d.) data, these sample size estimates can be used to construct a band around the empirical CDFs over $d_{\mathrm{nearest}}$ (Eq. 5) using

the sharp constant (Massart, 1990) of the distribution-free DKW inequality (Dvoretzky et al., 1956), with $\hat{n}_{\min}$ taken as the minimum among the estimated sample sizes across classes for the test instance:

$$\epsilon = \sqrt{\frac{1}{2 \cdot \hat{n}_{\min}} \log_e \left(\frac{2}{1-\alpha}\right)}, \tag{11}$$

$$\hat{n}_{\min} = \min\left[\hat{n}_1, \ldots, \hat{n}_C\right] \tag{12}$$

If $\hat{n}_{\min} = 0$, our convention is to set $\epsilon = 1$. We then construct the conservative lower and upper counterparts to the distance quantile of Eq. 5:

$$d_{\text{lower}} = \max\left(d - \epsilon, 0\right) \tag{13}$$

$$d_{\text{upper}} = \min\left(d + \epsilon, 1\right) \tag{14}$$

Eq. 6 can then be calculated by substituting $d_{\text{lower}}$ and $d_{\text{upper}}$, in turn, for $d$, resulting in a band around the prediction-conditional estimate.

### A.4 Comparison to Bayesian Last-layer Networks

In the main text, we compare to representative Frequentist and empirically motivated estimators. In this section, we further compare to variational Bayesian last-layer neural networks (VBLL), a computationally efficient Bayesian approach (Harrison et al., 2024). The basic setup is similar to the Frequentist and empirically motivated approaches examined in the main text in that it involves training a small final-layer adaptor over the frozen parameters of the language model. However, in this case, the adaptor network is a multi-layer perceptron (MLP) combined with the VBLL estimator. We consider both the discriminative and generative VBLL estimators.

**Models.** We follow the parameter choices and architecture of Harrison et al. (2024), and its associated code tutorial[8], with applicable changes to match the experimental settings of the other models and estimators. Specifically, the VBLL models consist of an input linear layer, 2 core linear layers, and a final output linear layer. The hidden dimension is set at 795, which yields a similar number of parameters as the SDM activation layers (approximately 6 million parameters for the `Phi-3.5-mini-instruct` adaptors and approximately 8 million parameters for the `Mixtral 8x7B Instruct v0.1` adaptors). The input to the VBLL models is the same mean-centered embeddings used with the SDM activation layers and the baseline CNN adaptor layers of the main text. We use the AdamW optimizer (Loshchilov & Hutter, 2019) with a weight decay of $1 \times 10^{-4}$, which following the code tutorial, is not applied to the final layer. We use a learning rate of $1 \times 10^{-5}$ for training, matching that used with the SDM activation layers. Following the code tutorial, we use gradient clipping with a max norm of 1, and we use the exponential linear unit (ELU) as the activation function (Clevert et al., 2016). We train for 200 epochs, choosing the epoch with the lowest held-out validation loss over $\mathcal{D}_{\text{ca}}$ as the chosen weights. We repeat this process for $J = 10$ shuffles of the data (i.e., splits of $\mathcal{D}_{\text{tr}}$ and $\mathcal{D}_{\text{ca}}$), choosing the model with the globally lowest overall held-out validation loss over $\mathcal{D}_{\text{ca}}$ as the final model.

For the discriminative models over `Phi-3.5-mini-instruct` and `Mixtral 8x7B Instruct v0.1`, we use the labels PHI3.5+DISCVBLLMLP and MIXTRAL8X7B+DISCVBLLMLP, respectively. For the generative models over `Phi-3.5-mini-instruct` and `Mixtral 8x7B Instruct v0.1`, we use the labels PHI3.5+GENVBLLMLP and MIXTRAL8X7B+GENVBLLMLP, respectively. For all of the aforementioned models, we use a KL regularization weight set at $\frac{1}{|\mathcal{D}_{\text{tr}}|}$, as in the original paper. We also consider analogous models that increase the KL regularization weight by a multiplicative factor of 50. Increasing the KL regularization weight is suggested in the code tutorial as the "simplest and most effective method to control the scale of uncertainty" of VBLL models. For these models with the larger KL regularization weight, we use the labels PHI3.5+DISCVBLLMLP$_{\text{rw50}}$, MIXTRAL8X7B+DISCVBLLMLP$_{\text{rw50}}$, PHI3.5+GENVBLLMLP$_{\text{rw50}}$, and MIXTRAL8X7B+GENVBLLMLP$_{\text{rw50}}$, respectively.

---

[8]Available at `https://github.com/VectorInstitute/vbll`

We use the label VBLL for the estimator that thresholds the output of the variational Bayesian last-layer neural network at $\alpha$ for the predicted class, analogous to the softmax and SDM estimators. The NO-REJECT estimator provides a reference point without any selection criteria applied (i.e., the standard marginal and class- and prediction-conditional accuracies over the given dataset).

**Results.** The results for the sentiment datasets appear in Table 6, and the results for the factcheck datasets appear in Table 7. For reference, in all cases we also provide the results over the held-out calibration set, $\mathcal{D}_{\text{ca}}$, which is the held-out split used to determine the final model weights, as noted above.

Comparing to Table 3 in the main text, the accuracies over $\mathcal{D}_{\text{ca}}$ for the NO-REJECT estimators are similar for the VBLL models and the models using SDM activation functions. The MLPs of the VBLL models and the 1-D CNNs of the SDM activation layers have a similar number of parameters and are trained with the same number of epochs and the same number of iterated shuffles of $\mathcal{D}_{\text{tr}}$ and $\mathcal{D}_{\text{ca}}$. The accuracies without selection for the SENTIMENT calibration set are already at least $\alpha$, and those for the FACTCHECK calibration set are below $\alpha$, but at least $\alpha - 0.05$. As such, differences in calibration effectiveness of the respective methods are not directly attributable to substantively different baseline accuracies for the in-distribution tasks.

We find that the VBLL estimators are well-calibrated in high-probability regions over in-distribution data, but generally fare poorly over co-variate shifts and out-of-distribution data. We find no clear advantages nor disadvantages for the discriminative vs. generative variants, and modifying the KL regularization weight has a minimal impact, at least at this scale. In Harrison et al. (2024), VBLL estimators over LMs for sentiment classification are only compared to an MLP baseline on in-distribution data; our evaluation setting is significantly more challenging and closer to real-world conditions encountered with LM applications.

These results corroborate the patterns observed in the main text. In our controlled examination of representative Bayesian, Frequentist, and empirically motivated estimators, the $\text{SDM}_{\text{HR}}$ estimator (i.e., the SDM activation function combined with the selection criteria of Eq. 9) is the only estimator that remains well-calibrated in high-probability regions across the in-distribution, co-variate shifted, and out-of-distribution data.

Table 4: Comparison of estimators for the underlined shuffled sentiment datasets, with $\alpha = 0.95$. R indicates all predictions were rejected, which is preferred over falling under the expected accuracy. $n = |\text{Admitted}|$, the count of non-rejected documents.

| Dataset | Model | Estimator | Class-conditional $y = 0$ Acc. | $\frac{n}{|\mathcal{D}_{te}|}$ | $y = 1$ Acc. | $\frac{n}{|\mathcal{D}_{te}|}$ | Prediction-conditional $\hat{y} = 0$ Acc. | $\frac{n}{|\mathcal{D}_{te}|}$ | $\hat{y} = 1$ Acc. | $\frac{n}{|\mathcal{D}_{te}|}$ | Marginal $y \in \{0,1\}$ Acc. | $\frac{n}{|\mathcal{D}_{te}|}$ |
|---|---|---|---|---|---|---|---|---|---|---|---|---|
| SENTIMENTSHUFFLED | PHI3.5 | NO-REJECT | 1.00 | 0.50 | 0.18 | 0.50 | 0.55 | 0.91 | 0.98 | 0.09 | 0.59 | 1. |
| SENTIMENTSHUFFLED | PHI3.5 | SOFTMAX | 1.00 | 0.50 | 0.16 | 0.45 | 0.57 | 0.88 | 0.99 | 0.07 | 0.60 | 0.95 |
| SENTIMENTSHUFFLED | PHI3.5 | TEMPSCALING | 1. | 0.44 | 0.12 | 0.18 | 0.74 | 0.60 | 1. | 0.02 | 0.75 | 0.62 |
| SENTIMENTSHUFFLED | PHI3.5 | APS | 1.00 | 0.42 | 0.14 | 0.15 | 0.76 | 0.55 | 0.97 | 0.02 | 0.77 | 0.57 |
| SENTIMENTSHUFFLED | PHI3.5 | RAPS | 1. | 0.41 | 0.13 | 0.16 | 0.75 | 0.55 | 1. | 0.02 | 0.76 | 0.57 |
| SENTIMENTSHUFFLED | PHI3.5+ADAPTOR | NO-REJECT | 0.83 | 0.50 | 0.81 | 0.50 | 0.81 | 0.51 | 0.82 | 0.49 | 0.82 | 1. |
| SENTIMENTSHUFFLED | PHI3.5+ADAPTOR | SOFTMAX | 0.98 | 0.13 | 0.97 | 0.12 | 0.98 | 0.13 | 0.97 | 0.12 | 0.97 | 0.25 |
| SENTIMENTSHUFFLED | PHI3.5+ADAPTOR | TEMPSCALING | 0.98 | 0.11 | 0.97 | 0.10 | 0.97 | 0.11 | 0.97 | 0.10 | 0.97 | 0.21 |
| SENTIMENTSHUFFLED | PHI3.5+ADAPTOR | APS | 0.93 | 0.23 | 0.90 | 0.24 | 0.90 | 0.24 | 0.93 | 0.23 | 0.91 | 0.48 |
| SENTIMENTSHUFFLED | PHI3.5+ADAPTOR | RAPS | 0.92 | 0.24 | 0.91 | 0.24 | 0.91 | 0.24 | 0.92 | 0.24 | 0.92 | 0.48 |
| SENTIMENTSHUFFLED | PHI3.5+SDM | NO-REJECT | 0.99 | 0.50 | 0.32 | 0.50 | 0.59 | 0.83 | 0.97 | 0.17 | 0.66 | 1. |
| SENTIMENTSHUFFLED | PHI3.5+SDM | SOFTMAX | 0.99 | 0.49 | 0.29 | 0.36 | 0.65 | 0.74 | 0.98 | 0.11 | 0.69 | 0.85 |
| SENTIMENTSHUFFLED | PHI3.5+SDM | SOFTMAX($d \cdot \mathbf{z}'$) | R | 0. | R | 0. | R | 0. | R | 0. | R | 0. |
| SENTIMENTSHUFFLED | PHI3.5+SDM | SDM | 1. | 0.01 | 1. | <0.01 | 1. | 0.01 | 1. | <0.01 | 1. | 0.01 |
| SENTIMENTSHUFFLED | PHI3.5+SDM | SDM$_{HR}$ | 1. | <0.01 | 1. | <0.01 | 1. | <0.01 | 1. | <0.01 | 1. | <0.01 |
| SENTIMENTSHUFFLED | MIXTRAL8x7B | NO-REJECT | 0.99 | 0.50 | 0.35 | 0.50 | 0.60 | 0.82 | 0.97 | 0.18 | 0.67 | 1. |
| SENTIMENTSHUFFLED | MIXTRAL8x7B | SOFTMAX | 0.99 | 0.50 | 0.34 | 0.48 | 0.61 | 0.81 | 0.97 | 0.17 | 0.67 | 0.98 |
| SENTIMENTSHUFFLED | MIXTRAL8x7B | TEMPSCALING | 0.99 | 0.48 | 0.37 | 0.38 | 0.67 | 0.72 | 0.98 | 0.14 | 0.72 | 0.86 |
| SENTIMENTSHUFFLED | MIXTRAL8x7B | APS | 0.99 | 0.45 | 0.44 | 0.27 | 0.75 | 0.60 | 0.97 | 0.12 | 0.79 | 0.72 |
| SENTIMENTSHUFFLED | MIXTRAL8x7B | RAPS | 0.99 | 0.44 | 0.43 | 0.29 | 0.73 | 0.60 | 0.98 | 0.13 | 0.77 | 0.73 |
| SENTIMENTSHUFFLED | MIXTRAL8x7B+ADAPTOR | NO-REJECT | 0.91 | 0.50 | 0.72 | 0.50 | 0.76 | 0.60 | 0.89 | 0.40 | 0.81 | 1. |
| SENTIMENTSHUFFLED | MIXTRAL8x7B+ADAPTOR | SOFTMAX | 0.99 | 0.28 | 0.83 | 0.14 | 0.92 | 0.31 | 0.98 | 0.12 | 0.94 | 0.43 |
| SENTIMENTSHUFFLED | MIXTRAL8x7B+ADAPTOR | TEMPSCALING | 0.99 | 0.26 | 0.82 | 0.11 | 0.93 | 0.28 | 0.98 | 0.09 | 0.94 | 0.37 |
| SENTIMENTSHUFFLED | MIXTRAL8x7B+ADAPTOR | APS | 0.97 | 0.34 | 0.78 | 0.24 | 0.86 | 0.38 | 0.95 | 0.19 | 0.89 | 0.58 |
| SENTIMENTSHUFFLED | MIXTRAL8x7B+ADAPTOR | RAPS | 0.96 | 0.35 | 0.79 | 0.23 | 0.87 | 0.38 | 0.93 | 0.20 | 0.89 | 0.58 |
| SENTIMENTSHUFFLED | MIXTRAL8x7B+SDM | NO-REJECT | 0.79 | 0.50 | 0.82 | 0.50 | 0.82 | 0.48 | 0.79 | 0.52 | 0.80 | 1. |
| SENTIMENTSHUFFLED | MIXTRAL8x7B+SDM | SOFTMAX | 0.80 | 0.48 | 0.83 | 0.49 | 0.82 | 0.47 | 0.80 | 0.50 | 0.81 | 0.97 |
| SENTIMENTSHUFFLED | MIXTRAL8x7B+SDM | SOFTMAX($d \cdot \mathbf{z}'$) | R | 0. | R | 0. | R | 0. | R | 0. | R | 0. |
| SENTIMENTSHUFFLED | MIXTRAL8x7B+SDM | SDM | 1. | <0.01 | R | 0. | 1. | <0.01 | R | 0. | 1. | <0.01 |
| SENTIMENTSHUFFLED | MIXTRAL8x7B+SDM | SDM$_{HR}$ | R | 0. | R | 0. | R | 0. | R | 0. | R | 0. |
| SENTIMENTOODSHUFFLED | PHI3.5 | NO-REJECT | 1.00 | 0.50 | 0.35 | 0.50 | 0.60 | 0.82 | 0.99 | 0.18 | 0.67 | 1. |
| SENTIMENTOODSHUFFLED | PHI3.5 | SOFTMAX | 1.00 | 0.50 | 0.34 | 0.47 | 0.62 | 0.81 | 0.99 | 0.16 | 0.68 | 0.97 |
| SENTIMENTOODSHUFFLED | PHI3.5 | TEMPSCALING | 1.00 | 0.49 | 0.34 | 0.29 | 0.72 | 0.68 | 0.99 | 0.10 | 0.75 | 0.78 |
| SENTIMENTOODSHUFFLED | PHI3.5 | APS | 1.00 | 0.48 | 0.36 | 0.27 | 0.74 | 0.65 | 0.99 | 0.10 | 0.77 | 0.75 |
| SENTIMENTOODSHUFFLED | PHI3.5 | RAPS | 1.00 | 0.49 | 0.36 | 0.27 | 0.74 | 0.66 | 0.99 | 0.10 | 0.77 | 0.76 |
| SENTIMENTOODSHUFFLED | PHI3.5+ADAPTOR | NO-REJECT | 0.64 | 0.50 | 0.64 | 0.50 | 0.64 | 0.50 | 0.64 | 0.50 | 0.64 | 1. |
| SENTIMENTOODSHUFFLED | PHI3.5+ADAPTOR | SOFTMAX | 0.78 | 0.01 | 0.93 | 0.03 | 0.81 | 0.01 | 0.92 | 0.03 | 0.89 | 0.04 |
| SENTIMENTOODSHUFFLED | PHI3.5+ADAPTOR | TEMPSCALING | 0.79 | 0.01 | 0.94 | 0.03 | 0.83 | 0.01 | 0.93 | 0.03 | 0.90 | 0.03 |
| SENTIMENTOODSHUFFLED | PHI3.5+ADAPTOR | APS | 0.68 | 0.11 | 0.73 | 0.14 | 0.67 | 0.12 | 0.74 | 0.14 | 0.71 | 0.26 |
| SENTIMENTOODSHUFFLED | PHI3.5+ADAPTOR | RAPS | 0.70 | 0.12 | 0.72 | 0.14 | 0.68 | 0.12 | 0.74 | 0.13 | 0.71 | 0.25 |
| SENTIMENTOODSHUFFLED | PHI3.5+SDM | NO-REJECT | 0.97 | 0.50 | 0.63 | 0.50 | 0.72 | 0.67 | 0.95 | 0.33 | 0.80 | 1. |
| SENTIMENTOODSHUFFLED | PHI3.5+SDM | SOFTMAX | 0.98 | 0.47 | 0.64 | 0.43 | 0.75 | 0.54 | 0.97 | 0.28 | 0.82 | 0.89 |
| SENTIMENTOODSHUFFLED | PHI3.5+SDM | SOFTMAX($d \cdot \mathbf{z}'$) | R | 0. | R | 0. | R | 0. | R | 0. | R | 0. |
| SENTIMENTOODSHUFFLED | PHI3.5+SDM | SDM | 1. | <0.01 | 1. | <0.01 | 1. | <0.01 | 1. | <0.01 | 1. | 0.01 |
| SENTIMENTOODSHUFFLED | PHI3.5+SDM | SDM$_{HR}$ | 1. | <0.01 | 1. | <0.01 | 1. | <0.01 | 1. | <0.01 | 1. | <0.01 |
| SENTIMENTOODSHUFFLED | MIXTRAL8x7B | NO-REJECT | 1.00 | 0.50 | 0.12 | 0.50 | 0.53 | 0.94 | 1.00 | 0.06 | 0.56 | 1. |
| SENTIMENTOODSHUFFLED | MIXTRAL8x7B | SOFTMAX | 1.00 | 0.50 | 0.12 | 0.49 | 0.54 | 0.93 | 1.00 | 0.06 | 0.56 | 0.99 |
| SENTIMENTOODSHUFFLED | MIXTRAL8x7B | TEMPSCALING | 1. | 0.49 | 0.14 | 0.34 | 0.63 | 0.78 | 1. | 0.05 | 0.65 | 0.83 |
| SENTIMENTOODSHUFFLED | MIXTRAL8x7B | APS | 1. | 0.36 | 0.24 | 0.18 | 0.72 | 0.51 | 1. | 0.04 | 0.74 | 0.55 |
| SENTIMENTOODSHUFFLED | MIXTRAL8x7B | RAPS | 1. | 0.36 | 0.23 | 0.19 | 0.72 | 0.51 | 1. | 0.04 | 0.74 | 0.55 |
| SENTIMENTOODSHUFFLED | MIXTRAL8x7B+ADAPTOR | NO-REJECT | 0.97 | 0.50 | 0.25 | 0.50 | 0.56 | 0.86 | 0.89 | 0.14 | 0.61 | 1. |
| SENTIMENTOODSHUFFLED | MIXTRAL8x7B+ADAPTOR | SOFTMAX | 1. | 0.04 | 0.23 | 0.03 | 0.62 | 0.07 | 1. | 0.01 | 0.66 | 0.07 |
| SENTIMENTOODSHUFFLED | MIXTRAL8x7B+ADAPTOR | TEMPSCALING | 1. | 0.02 | 0.27 | 0.02 | 0.58 | 0.03 | 1. | 0.01 | 0.64 | 0.04 |
| SENTIMENTOODSHUFFLED | MIXTRAL8x7B+ADAPTOR | APS | 0.99 | 0.18 | 0.21 | 0.15 | 0.60 | 0.29 | 0.94 | 0.03 | 0.64 | 0.32 |
| SENTIMENTOODSHUFFLED | MIXTRAL8x7B+ADAPTOR | RAPS | 0.99 | 0.18 | 0.21 | 0.14 | 0.61 | 0.29 | 0.93 | 0.03 | 0.64 | 0.32 |
| SENTIMENTOODSHUFFLED | MIXTRAL8x7B+SDM | NO-REJECT | 0.75 | 0.50 | 0.71 | 0.50 | 0.72 | 0.52 | 0.74 | 0.48 | 0.73 | 1. |
| SENTIMENTOODSHUFFLED | MIXTRAL8x7B+SDM | SOFTMAX | 0.79 | 0.43 | 0.73 | 0.45 | 0.73 | 0.46 | 0.78 | 0.42 | 0.76 | 0.89 |
| SENTIMENTOODSHUFFLED | MIXTRAL8x7B+SDM | SOFTMAX($d \cdot \mathbf{z}'$) | 1. | <0.01 | 1. | <0.01 | 1. | <0.01 | 1. | <0.01 | 1. | <0.01 |
| SENTIMENTOODSHUFFLED | MIXTRAL8x7B+SDM | SDM | 1. | <0.01 | 0.83 | <0.01 | 0.88 | 0.01 | 1. | <0.01 | 0.93 | 0.01 |
| SENTIMENTOODSHUFFLED | MIXTRAL8x7B+SDM | SDM$_{HR}$ | 1. | <0.01 | 1. | <0.01 | 1. | <0.01 | 1. | <0.01 | 1. | <0.01 |

Table 5: Comparison of estimators for the underlined shuffled factcheck datasets. Unless specified otherwise, $\alpha = 0.95$. R indicates all predictions were rejected, which is preferred over falling under the expected accuracy. $n = |\text{Admitted}|$, the count of non-rejected documents.

| Dataset | Model | Estimator | Class-conditional $y=0$ Acc. | $\frac{n}{|\mathcal{D}_{\text{te}}|}$ | $y=1$ Acc. | $\frac{n}{|\mathcal{D}_{\text{te}}|}$ | Prediction-conditional $\hat{y}=0$ Acc. | $\frac{n}{|\mathcal{D}_{\text{te}}|}$ | $\hat{y}=1$ Acc. | $\frac{n}{|\mathcal{D}_{\text{te}}|}$ | Marginal $y\in\{0,1\}$ Acc. | $\frac{n}{|\mathcal{D}_{\text{te}}|}$ |
|---|---|---|---|---|---|---|---|---|---|---|---|---|
| FACTCHECKSHUFFLED | PHI3.5 | NO-REJECT | 0.91 | 1. | - | 0. | 1. | 0.91 | 0. | 0.09 | 0.91 | 1. |
| FACTCHECKSHUFFLED | PHI3.5 | SOFTMAX | 0.92 | 0.99 | - | 0. | 1. | 0.91 | 0. | 0.08 | 0.92 | 0.99 |
| FACTCHECKSHUFFLED | PHI3.5 | TEMPSCALING | 0.93 | 0.87 | - | 0. | 1. | 0.81 | 0. | 0.06 | 0.93 | 0.87 |
| FACTCHECKSHUFFLED | PHI3.5 | APS | 0.93 | 0.45 | - | 0. | 1. | 0.42 | 0. | 0.03 | 0.93 | 0.45 |
| FACTCHECKSHUFFLED | PHI3.5 | RAPS | 0.95 | 0.52 | - | 0. | 1. | 0.50 | 0. | 0.02 | 0.95 | 0.52 |
| FACTCHECKSHUFFLED | PHI3.5+ADAPTOR | NO-REJECT | 0.34 | 1. | - | 0. | 1. | 0.34 | 0. | 0.66 | 0.34 | 1. |
| FACTCHECKSHUFFLED | PHI3.5+ADAPTOR | SOFTMAX | 0.20 | 0.24 | - | 0. | 1. | 0.05 | 0. | 0.19 | 0.20 | 0.24 |
| FACTCHECKSHUFFLED | PHI3.5+ADAPTOR | TEMPSCALING | 0.13 | 0.19 | - | 0. | 1. | 0.02 | 0. | 0.17 | 0.13 | 0.19 |
| FACTCHECKSHUFFLED | PHI3.5+ADAPTOR | APS | 0.24 | 0.38 | - | 0. | 1. | 0.09 | 0. | 0.29 | 0.24 | 0.38 |
| FACTCHECKSHUFFLED | PHI3.5+ADAPTOR | RAPS | 0.27 | 0.39 | - | 0. | 1. | 0.11 | 0. | 0.29 | 0.27 | 0.39 |
| FACTCHECKSHUFFLED | PHI3.5+SDM | NO-REJECT | 0.66 | 1. | - | 0. | 1. | 0.66 | 0. | 0.34 | 0.66 | 1. |
| FACTCHECKSHUFFLED | PHI3.5+SDM | SOFTMAX | 0.69 | 0.64 | - | 0. | 1. | 0.44 | 0. | 0.20 | 0.69 | 0.64 |
| FACTCHECKSHUFFLED | PHI3.5+SDM | SOFTMAX$(d\cdot\boldsymbol{z}')$ | R | 0. | - | 0. | R | 0. | R | 0. | R | 0. |
| FACTCHECKSHUFFLED | PHI3.5+SDM | SDM | R | 0. | - | 0. | R | 0. | R | 0. | R | 0. |
| FACTCHECKSHUFFLED | PHI3.5+SDM | SDM$_{\text{HR}}$ | R | 0. | - | 0. | R | 0. | R | 0. | R | 0. |
| FACTCHECKSHUFFLED | MIXTRAL8X7B | NO-REJECT | 0.98 | 1. | - | 0. | 1. | 0.98 | 0. | 0.02 | 0.98 | 1. |
| FACTCHECKSHUFFLED | MIXTRAL8X7B | SOFTMAX | 0.98 | 1. | - | 0. | 1. | 0.98 | 0. | 0.02 | 0.98 | 1. |
| FACTCHECKSHUFFLED | MIXTRAL8X7B | TEMPSCALING | 0.98 | 0.98 | - | 0. | 1. | 0.96 | 0. | 0.02 | 0.98 | 0.98 |
| FACTCHECKSHUFFLED | MIXTRAL8X7B | APS | 0.98 | 0.18 | - | 0. | 1. | 0.18 | 0. | <0.01 | 0.98 | 0.18 |
| FACTCHECKSHUFFLED | MIXTRAL8X7B | RAPS | 0.98 | 0.23 | - | 0. | 1. | 0.23 | 0. | <0.01 | 0.98 | 0.23 |
| FACTCHECKSHUFFLED | MIXTRAL8X7B+ADAPTOR | NO-REJECT | 0.79 | 1. | - | 0. | 1. | 0.79 | 0. | 0.21 | 0.79 | 1. |
| FACTCHECKSHUFFLED | MIXTRAL8X7B+ADAPTOR | SOFTMAX | 0.69 | 0.13 | - | 0. | 1. | 0.09 | 0. | 0.04 | 0.69 | 0.13 |
| FACTCHECKSHUFFLED | MIXTRAL8X7B+ADAPTOR | TEMPSCALING | 0.55 | 0.09 | - | 0. | 1. | 0.05 | 0. | 0.04 | 0.55 | 0.09 |
| FACTCHECKSHUFFLED | MIXTRAL8X7B+ADAPTOR | APS | 0.77 | 0.40 | - | 0. | 1. | 0.31 | 0. | 0.09 | 0.77 | 0.40 |
| FACTCHECKSHUFFLED | MIXTRAL8X7B+ADAPTOR | RAPS | 0.79 | 0.39 | - | 0. | 1. | 0.31 | 0. | 0.08 | 0.79 | 0.39 |
| FACTCHECKSHUFFLED | MIXTRAL8X7B+SDM | NO-REJECT | 0.76 | 1. | - | 0. | 1. | 0.76 | 0. | 0.24 | 0.76 | 1. |
| FACTCHECKSHUFFLED | MIXTRAL8X7B+SDM | SOFTMAX | 0.79 | 0.65 | - | 0. | 1. | 0.51 | 0. | 0.13 | 0.79 | 0.65 |
| FACTCHECKSHUFFLED | MIXTRAL8X7B+SDM | SOFTMAX$(d\cdot\boldsymbol{z}')$ | R | 0. | - | 0. | R | 0. | R | 0. | R | 0. |
| FACTCHECKSHUFFLED | MIXTRAL8X7B+SDM | SDM | 1. | 0.01 | - | 0. | 1. | 0.01 | R | 0. | 1. | 0.01 |
| FACTCHECKSHUFFLED | MIXTRAL8X7B+SDM | SDM$_{\text{HR}}$ | R | 0. | - | 0. | R | 0. | R | 0. | R | 0. |
| FACTCHECKSHUFFLED | MIXTRAL8X7B+SDM | SDM, $\alpha=0.94$ | 1. | 0.01 | - | 0. | 1. | 0.01 | R | 0. | 1. | 0.01 |
| FACTCHECKSHUFFLED | MIXTRAL8X7B+SDM | SDM$_{\text{HR}}$, $\alpha=0.94$ | 1. | 0.01 | - | 0. | 1. | 0.01 | R | 0. | 1. | 0.01 |

Table 6: Comparison of Bayesian last-layer estimators (Harrison et al., 2024) for the sentiment datasets, including the shuffled challenge sets, with $\alpha = 0.95$. R indicates all predictions were rejected, which is preferred over falling under the expected accuracy. $n = |\text{Admitted}|$, the count of non-rejected documents.

| | | | Class-conditional | | | | Prediction-conditional | | | | Marginal | |
| | | | $y=0$ | | $y=1$ | | $\hat{y}=0$ | | $\hat{y}=1$ | | $y\in\{0,1\}$ | |
| Dataset | Model | Estimator | Acc. | $\frac{n}{|\mathcal{D}_{te}|}$ | Acc. | $\frac{n}{|\mathcal{D}_{te}|}$ | Acc. | $\frac{n}{|\mathcal{D}_{te}|}$ | Acc. | $\frac{n}{|\mathcal{D}_{te}|}$ | Acc. | $\frac{n}{|\mathcal{D}_{te}|}$ |
|---|---|---|---|---|---|---|---|---|---|---|---|---|
| Sentiment $\mathcal{D}_{ca}$ | PHI3.5+DiscVBLLMLP | NO-REJECT | 0.97 | 0.51 | 0.96 | 0.49 | 0.96 | 0.51 | 0.96 | 0.49 | 0.96 | 1. |
| Sentiment | PHI3.5+DiscVBLLMLP | NO-REJECT | 0.97 | 0.50 | 0.95 | 0.50 | 0.95 | 0.51 | 0.97 | 0.49 | 0.96 | 1. |
| SentimentOOD | PHI3.5+DiscVBLLMLP | NO-REJECT | 0.61 | 0.50 | 0.67 | 0.50 | 0.65 | 0.47 | 0.63 | 0.53 | 0.64 | 1. |
| SentimentShuffled | PHI3.5+DiscVBLLMLP | NO-REJECT | 0.87 | 0.50 | 0.75 | 0.50 | 0.78 | 0.56 | 0.85 | 0.44 | 0.81 | 1. |
| SentimentOODShuffled | PHI3.5+DiscVBLLMLP | NO-REJECT | 0.77 | 0.50 | 0.56 | 0.50 | 0.64 | 0.61 | 0.71 | 0.39 | 0.67 | 1. |
| Sentiment $\mathcal{D}_{ca}$ | PHI3.5+DiscVBLLMLP | VBLL | 0.99 | 0.42 | 0.99 | 0.40 | 0.99 | 0.42 | 0.99 | 0.40 | 0.99 | 0.82 |
| Sentiment | PHI3.5+DiscVBLLMLP | VBLL | 0.99 | 0.42 | 1.00 | 0.40 | 1.00 | 0.42 | 0.99 | 0.40 | 0.99 | 0.82 |
| SentimentOOD | PHI3.5+DiscVBLLMLP | VBLL | 0.89 | 0.03 | 0.96 | 0.04 | 0.93 | 0.02 | 0.94 | 0.05 | 0.94 | 0.07 |
| SentimentShuffled | PHI3.5+DiscVBLLMLP | VBLL | 0.99 | 0.13 | 0.95 | 0.05 | 0.98 | 0.13 | 0.97 | 0.05 | 0.98 | 0.18 |
| SentimentOODShuffled | PHI3.5+DiscVBLLMLP | VBLL | 0.98 | 0.02 | 0.86 | 0.02 | 0.89 | 0.02 | 0.97 | 0.01 | 0.92 | 0.04 |
| Sentiment $\mathcal{D}_{ca}$ | PHI3.5+DiscVBLLMLP$_{rw50}$ | NO-REJECT | 0.96 | 0.51 | 0.96 | 0.49 | 0.96 | 0.51 | 0.96 | 0.49 | 0.96 | 1. |
| Sentiment | PHI3.5+DiscVBLLMLP$_{rw50}$ | NO-REJECT | 0.97 | 0.50 | 0.95 | 0.50 | 0.95 | 0.51 | 0.97 | 0.49 | 0.96 | 1. |
| SentimentOOD | PHI3.5+DiscVBLLMLP$_{rw50}$ | NO-REJECT | 0.64 | 0.50 | 0.71 | 0.50 | 0.69 | 0.47 | 0.66 | 0.53 | 0.67 | 1. |
| SentimentShuffled | PHI3.5+DiscVBLLMLP$_{rw50}$ | NO-REJECT | 0.85 | 0.50 | 0.77 | 0.50 | 0.79 | 0.54 | 0.84 | 0.46 | 0.81 | 1. |
| SentimentOODShuffled | PHI3.5+DiscVBLLMLP$_{rw50}$ | NO-REJECT | 0.80 | 0.50 | 0.60 | 0.50 | 0.67 | 0.60 | 0.75 | 0.40 | 0.70 | 1. |
| Sentiment $\mathcal{D}_{ca}$ | PHI3.5+DiscVBLLMLP$_{rw50}$ | VBLL | 0.99 | 0.43 | 0.99 | 0.41 | 0.99 | 0.42 | 0.99 | 0.41 | 0.99 | 0.84 |
| Sentiment | PHI3.5+DiscVBLLMLP$_{rw50}$ | VBLL | 0.99 | 0.42 | 1.00 | 0.41 | 1.00 | 0.42 | 0.99 | 0.41 | 0.99 | 0.83 |
| SentimentOOD | PHI3.5+DiscVBLLMLP$_{rw50}$ | VBLL | 0.95 | 0.02 | 0.96 | 0.04 | 0.92 | 0.02 | 0.98 | 0.04 | 0.96 | 0.07 |
| SentimentShuffled | PHI3.5+DiscVBLLMLP$_{rw50}$ | VBLL | 0.99 | 0.13 | 0.96 | 0.09 | 0.98 | 0.13 | 0.98 | 0.09 | 0.98 | 0.22 |
| SentimentOODShuffled | PHI3.5+DiscVBLLMLP$_{rw50}$ | VBLL | 0.98 | 0.02 | 0.86 | 0.02 | 0.90 | 0.02 | 0.97 | 0.02 | 0.93 | 0.04 |
| Sentiment $\mathcal{D}_{ca}$ | PHI3.5+GenVBLLMLP | NO-REJECT | 0.96 | 0.50 | 0.96 | 0.50 | 0.96 | 0.50 | 0.96 | 0.50 | 0.96 | 1. |
| Sentiment | PHI3.5+GenVBLLMLP | NO-REJECT | 0.97 | 0.50 | 0.96 | 0.50 | 0.96 | 0.51 | 0.97 | 0.49 | 0.97 | 1. |
| SentimentOOD | PHI3.5+GenVBLLMLP | NO-REJECT | 0.28 | 0.50 | 0.93 | 0.50 | 0.80 | 0.17 | 0.56 | 0.83 | 0.60 | 1. |
| SentimentShuffled | PHI3.5+GenVBLLMLP | NO-REJECT | 0.94 | 0.50 | 0.60 | 0.50 | 0.70 | 0.67 | 0.91 | 0.33 | 0.77 | 1. |
| SentimentOODShuffled | PHI3.5+GenVBLLMLP | NO-REJECT | 0.43 | 0.50 | 0.85 | 0.50 | 0.74 | 0.29 | 0.60 | 0.71 | 0.64 | 1. |
| Sentiment $\mathcal{D}_{ca}$ | PHI3.5+GenVBLLMLP | VBLL | 0.99 | 0.43 | 0.99 | 0.44 | 0.99 | 0.43 | 0.99 | 0.44 | 0.99 | 0.87 |
| Sentiment | PHI3.5+GenVBLLMLP | VBLL | 0.99 | 0.44 | 0.99 | 0.43 | 0.99 | 0.44 | 0.99 | 0.43 | 0.99 | 0.87 |
| SentimentOOD | PHI3.5+GenVBLLMLP | VBLL | 0.13 | 0.16 | 0.99 | 0.28 | 0.89 | 0.02 | 0.67 | 0.41 | 0.68 | 0.43 |
| SentimentShuffled | PHI3.5+GenVBLLMLP | VBLL | 1.00 | 0.31 | 0.70 | 0.14 | 0.88 | 0.35 | 0.99 | 0.10 | 0.90 | 0.45 |
| SentimentOODShuffled | PHI3.5+GenVBLLMLP | VBLL | 0.32 | 0.08 | 0.97 | 0.19 | 0.80 | 0.03 | 0.77 | 0.24 | 0.78 | 0.27 |
| Sentiment $\mathcal{D}_{ca}$ | PHI3.5+GenVBLLMLP$_{rw50}$ | NO-REJECT | 0.96 | 0.51 | 0.97 | 0.49 | 0.97 | 0.50 | 0.96 | 0.50 | 0.96 | 1. |
| Sentiment | PHI3.5+GenVBLLMLP$_{rw50}$ | NO-REJECT | 0.96 | 0.50 | 0.96 | 0.50 | 0.96 | 0.50 | 0.96 | 0.50 | 0.96 | 1. |
| SentimentOOD | PHI3.5+GenVBLLMLP$_{rw50}$ | NO-REJECT | 0.43 | 0.50 | 0.78 | 0.50 | 0.66 | 0.32 | 0.58 | 0.68 | 0.61 | 1. |
| SentimentShuffled | PHI3.5+GenVBLLMLP$_{rw50}$ | NO-REJECT | 0.78 | 0.50 | 0.84 | 0.50 | 0.83 | 0.47 | 0.79 | 0.53 | 0.81 | 1. |
| SentimentOODShuffled | PHI3.5+GenVBLLMLP$_{rw50}$ | NO-REJECT | 0.64 | 0.50 | 0.76 | 0.50 | 0.72 | 0.44 | 0.68 | 0.56 | 0.70 | 1. |
| Sentiment $\mathcal{D}_{ca}$ | PHI3.5+GenVBLLMLP$_{rw50}$ | VBLL | 0.99 | 0.45 | 0.99 | 0.43 | 0.99 | 0.45 | 0.99 | 0.43 | 0.99 | 0.88 |
| Sentiment | PHI3.5+GenVBLLMLP$_{rw50}$ | VBLL | 0.99 | 0.44 | 0.99 | 0.44 | 0.99 | 0.44 | 0.99 | 0.44 | 0.99 | 0.89 |
| SentimentOOD | PHI3.5+GenVBLLMLP$_{rw50}$ | VBLL | 0.36 | 0.12 | 0.94 | 0.15 | 0.83 | 0.05 | 0.65 | 0.22 | 0.68 | 0.27 |
| SentimentShuffled | PHI3.5+GenVBLLMLP$_{rw50}$ | VBLL | 0.91 | 0.21 | 0.96 | 0.26 | 0.95 | 0.20 | 0.93 | 0.27 | 0.94 | 0.47 |
| SentimentOODShuffled | PHI3.5+GenVBLLMLP$_{rw50}$ | VBLL | 0.76 | 0.09 | 0.86 | 0.14 | 0.78 | 0.09 | 0.85 | 0.14 | 0.82 | 0.24 |
| Sentiment $\mathcal{D}_{ca}$ | Mixtral8x7B+DiscVBLLMLP | NO-REJECT | 0.96 | 0.50 | 0.96 | 0.50 | 0.96 | 0.50 | 0.96 | 0.50 | 0.96 | 1. |
| Sentiment | Mixtral8x7B+DiscVBLLMLP | NO-REJECT | 0.97 | 0.50 | 0.96 | 0.50 | 0.96 | 0.50 | 0.97 | 0.50 | 0.97 | 1. |
| SentimentOOD | Mixtral8x7B+DiscVBLLMLP | NO-REJECT | 0.84 | 0.50 | 0.60 | 0.50 | 0.68 | 0.62 | 0.79 | 0.38 | 0.72 | 1. |
| SentimentShuffled | Mixtral8x7B+DiscVBLLMLP | NO-REJECT | 0.89 | 0.50 | 0.78 | 0.50 | 0.80 | 0.55 | 0.87 | 0.45 | 0.83 | 1. |
| SentimentOODShuffled | Mixtral8x7B+DiscVBLLMLP | NO-REJECT | 0.91 | 0.50 | 0.40 | 0.50 | 0.60 | 0.76 | 0.82 | 0.24 | 0.66 | 1. |
| Sentiment $\mathcal{D}_{ca}$ | Mixtral8x7B+DiscVBLLMLP | VBLL | 0.99 | 0.44 | 0.99 | 0.44 | 0.99 | 0.44 | 0.99 | 0.44 | 0.99 | 0.88 |
| Sentiment | Mixtral8x7B+DiscVBLLMLP | VBLL | 0.99 | 0.44 | 0.99 | 0.43 | 0.99 | 0.44 | 0.99 | 0.43 | 0.99 | 0.87 |
| SentimentOOD | Mixtral8x7B+DiscVBLLMLP | VBLL | 0.94 | 0.01 | 0.97 | 0.11 | 0.79 | 0.02 | 0.99 | 0.10 | 0.96 | 0.12 |
| SentimentShuffled | Mixtral8x7B+DiscVBLLMLP | VBLL | 0.99 | 0.24 | 0.91 | 0.12 | 0.95 | 0.25 | 0.98 | 0.11 | 0.96 | 0.36 |
| SentimentOODShuffled | Mixtral8x7B+DiscVBLLMLP | VBLL | 1. | 0.01 | 0.82 | 0.01 | 0.68 | 0.01 | 1. | 0.01 | 0.87 | 0.02 |
| Sentiment $\mathcal{D}_{ca}$ | Mixtral8x7B+DiscVBLLMLP$_{rw50}$ | NO-REJECT | 0.96 | 0.50 | 0.97 | 0.50 | 0.97 | 0.50 | 0.96 | 0.50 | 0.96 | 1. |
| Sentiment | Mixtral8x7B+DiscVBLLMLP$_{rw50}$ | NO-REJECT | 0.97 | 0.50 | 0.97 | 0.50 | 0.97 | 0.50 | 0.97 | 0.50 | 0.97 | 1. |
| SentimentOOD | Mixtral8x7B+DiscVBLLMLP$_{rw50}$ | NO-REJECT | 0.89 | 0.50 | 0.59 | 0.50 | 0.69 | 0.65 | 0.84 | 0.35 | 0.74 | 1. |
| SentimentShuffled | Mixtral8x7B+DiscVBLLMLP$_{rw50}$ | NO-REJECT | 0.86 | 0.50 | 0.81 | 0.50 | 0.81 | 0.53 | 0.85 | 0.47 | 0.83 | 1. |
| SentimentOODShuffled | Mixtral8x7B+DiscVBLLMLP$_{rw50}$ | NO-REJECT | 0.94 | 0.50 | 0.44 | 0.50 | 0.63 | 0.75 | 0.88 | 0.25 | 0.69 | 1. |
| Sentiment $\mathcal{D}_{ca}$ | Mixtral8x7B+DiscVBLLMLP$_{rw50}$ | VBLL | 0.99 | 0.44 | 0.99 | 0.43 | 0.99 | 0.44 | 0.99 | 0.43 | 0.99 | 0.87 |
| Sentiment | Mixtral8x7B+DiscVBLLMLP$_{rw50}$ | VBLL | 0.99 | 0.44 | 0.99 | 0.43 | 0.99 | 0.44 | 0.99 | 0.43 | 0.99 | 0.87 |
| SentimentOOD | Mixtral8x7B+DiscVBLLMLP$_{rw50}$ | VBLL | 0.98 | 0.03 | 0.93 | 0.07 | 0.87 | 0.04 | 0.99 | 0.06 | 0.95 | 0.10 |
| SentimentShuffled | Mixtral8x7B+DiscVBLLMLP$_{rw50}$ | VBLL | 0.99 | 0.24 | 0.92 | 0.14 | 0.95 | 0.25 | 0.98 | 0.13 | 0.96 | 0.38 |
| SentimentOODShuffled | Mixtral8x7B+DiscVBLLMLP$_{rw50}$ | VBLL | 1. | 0.02 | 0.69 | 0.01 | 0.87 | 0.02 | 1. | 0.01 | 0.90 | 0.03 |
| Sentiment $\mathcal{D}_{ca}$ | Mixtral8x7B+GenVBLLMLP | NO-REJECT | 0.97 | 0.50 | 0.97 | 0.50 | 0.97 | 0.50 | 0.97 | 0.50 | 0.97 | 1. |
| Sentiment | Mixtral8x7B+GenVBLLMLP | NO-REJECT | 0.97 | 0.50 | 0.96 | 0.50 | 0.96 | 0.51 | 0.97 | 0.49 | 0.97 | 1. |
| SentimentOOD | Mixtral8x7B+GenVBLLMLP | NO-REJECT | 0.80 | 0.50 | 0.72 | 0.50 | 0.74 | 0.54 | 0.78 | 0.46 | 0.76 | 1. |
| SentimentShuffled | Mixtral8x7B+GenVBLLMLP | NO-REJECT | 0.84 | 0.50 | 0.82 | 0.50 | 0.82 | 0.51 | 0.83 | 0.49 | 0.83 | 1. |
| SentimentOODShuffled | Mixtral8x7B+GenVBLLMLP | NO-REJECT | 0.85 | 0.50 | 0.54 | 0.50 | 0.65 | 0.65 | 0.78 | 0.35 | 0.69 | 1. |
| Sentiment $\mathcal{D}_{ca}$ | Mixtral8x7B+GenVBLLMLP | VBLL | 0.99 | 0.44 | 0.99 | 0.45 | 0.99 | 0.44 | 0.99 | 0.46 | 0.99 | 0.89 |
| Sentiment | Mixtral8x7B+GenVBLLMLP | VBLL | 0.99 | 0.44 | 0.99 | 0.44 | 0.99 | 0.44 | 0.99 | 0.45 | 0.99 | 0.88 |
| SentimentOOD | Mixtral8x7B+GenVBLLMLP | VBLL | 0.93 | 0.04 | 0.99 | 0.15 | 0.95 | 0.04 | 0.98 | 0.15 | 0.97 | 0.19 |
| SentimentShuffled | Mixtral8x7B+GenVBLLMLP | VBLL | 0.96 | 0.23 | 0.91 | 0.20 | 0.93 | 0.24 | 0.95 | 0.20 | 0.94 | 0.44 |
| SentimentOODShuffled | Mixtral8x7B+GenVBLLMLP | VBLL | 0.99 | 0.02 | 0.81 | 0.04 | 0.73 | 0.03 | 0.99 | 0.03 | 0.87 | 0.06 |
| Sentiment $\mathcal{D}_{ca}$ | Mixtral8x7B+GenVBLLMLP$_{rw50}$ | NO-REJECT | 0.97 | 0.50 | 0.97 | 0.50 | 0.97 | 0.50 | 0.97 | 0.50 | 0.97 | 1. |
| Sentiment | Mixtral8x7B+GenVBLLMLP$_{rw50}$ | NO-REJECT | 0.97 | 0.50 | 0.96 | 0.50 | 0.96 | 0.50 | 0.97 | 0.50 | 0.97 | 1. |
| SentimentOOD | Mixtral8x7B+GenVBLLMLP$_{rw50}$ | NO-REJECT | 0.79 | 0.50 | 0.73 | 0.50 | 0.75 | 0.53 | 0.78 | 0.47 | 0.76 | 1. |
| SentimentShuffled | Mixtral8x7B+GenVBLLMLP$_{rw50}$ | NO-REJECT | 0.83 | 0.50 | 0.82 | 0.50 | 0.82 | 0.51 | 0.83 | 0.49 | 0.83 | 1. |
| SentimentOODShuffled | Mixtral8x7B+GenVBLLMLP$_{rw50}$ | NO-REJECT | 0.83 | 0.50 | 0.56 | 0.50 | 0.65 | 0.64 | 0.77 | 0.36 | 0.69 | 1. |
| Sentiment $\mathcal{D}_{ca}$ | Mixtral8x7B+GenVBLLMLP$_{rw50}$ | VBLL | 0.98 | 0.44 | 0.99 | 0.46 | 0.99 | 0.44 | 0.98 | 0.46 | 0.99 | 0.90 |
| Sentiment | Mixtral8x7B+GenVBLLMLP$_{rw50}$ | VBLL | 0.99 | 0.44 | 0.99 | 0.45 | 0.99 | 0.44 | 0.99 | 0.45 | 0.99 | 0.89 |
| SentimentOOD | Mixtral8x7B+GenVBLLMLP$_{rw50}$ | VBLL | 0.91 | 0.04 | 0.99 | 0.16 | 0.95 | 0.04 | 0.98 | 0.16 | 0.97 | 0.20 |
| SentimentShuffled | Mixtral8x7B+GenVBLLMLP$_{rw50}$ | VBLL | 0.95 | 0.24 | 0.91 | 0.21 | 0.92 | 0.24 | 0.95 | 0.21 | 0.94 | 0.45 |
| SentimentOODShuffled | Mixtral8x7B+GenVBLLMLP$_{rw50}$ | VBLL | 0.98 | 0.02 | 0.82 | 0.04 | 0.73 | 0.03 | 0.99 | 0.04 | 0.88 | 0.06 |

Table 7: Comparison of Bayesian last-layer estimators (Harrison et al., 2024) for the factcheck datasets, including the shuffled challenge sets, with $\alpha = 0.95$. R indicates all predictions were rejected, which is preferred over falling under the expected accuracy. $n = |\text{Admitted}|$, the count of non-rejected documents.

| | | | Class-conditional | | | | Prediction-conditional | | | | Marginal | |
| | | | $y=0$ | | $y=1$ | | $\hat{y}=0$ | | $\hat{y}=1$ | | $y \in \{0,1\}$ | |
| Dataset | Model | Estimator | Acc. | $\frac{n}{|\mathcal{D}_{te}|}$ | Acc. | $\frac{n}{|\mathcal{D}_{te}|}$ | Acc. | $\frac{n}{|\mathcal{D}_{te}|}$ | Acc. | $\frac{n}{|\mathcal{D}_{te}|}$ | Acc. | $\frac{n}{|\mathcal{D}_{te}|}$ |
|---|---|---|---|---|---|---|---|---|---|---|---|---|
| FACTCHECK $\mathcal{D}_{ca}$ | PHI3.5+DiscVBLLMLP | NO-REJECT | 0.92 | 0.50 | 0.91 | 0.50 | 0.91 | 0.50 | 0.92 | 0.50 | 0.91 | 1. |
| FACTCHECK | PHI3.5+DiscVBLLMLP | NO-REJECT | 0.38 | 0.51 | 0.92 | 0.49 | 0.84 | 0.23 | 0.59 | 0.77 | 0.64 | 1. |
| FACTCHECKSHUFFLED | PHI3.5+DiscVBLLMLP | NO-REJECT | 0.33 | 1. | R | 0. | 1. | 0.33 | 0. | 0.67 | 0.33 | 1. |
| FACTCHECK $\mathcal{D}_{ca}$ | PHI3.5+DiscVBLLMLP | VBLL | 0.98 | 0.32 | 0.99 | 0.28 | 0.99 | 0.32 | 0.98 | 0.29 | 0.98 | 0.61 |
| FACTCHECK | PHI3.5+DiscVBLLMLP | VBLL | 0.35 | 0.07 | 1. | 0.31 | 1. | 0.02 | 0.88 | 0.36 | 0.88 | 0.38 |
| FACTCHECKSHUFFLED | PHI3.5+DiscVBLLMLP | VBLL | 0.15 | 0.21 | R | 0. | 1. | 0.03 | 0. | 0.18 | 0.15 | 0.21 |
| FACTCHECK $\mathcal{D}_{ca}$ | PHI3.5+DiscVBLLMLP$_{rw50}$ | NO-REJECT | 0.91 | 0.50 | 0.92 | 0.50 | 0.92 | 0.49 | 0.91 | 0.51 | 0.91 | 1. |
| FACTCHECK | PHI3.5+DiscVBLLMLP$_{rw50}$ | NO-REJECT | 0.45 | 0.51 | 0.94 | 0.49 | 0.89 | 0.26 | 0.62 | 0.74 | 0.69 | 1. |
| FACTCHECKSHUFFLED | PHI3.5+DiscVBLLMLP$_{rw50}$ | NO-REJECT | 0.43 | 1. | R | 0. | 1. | 0.43 | 0. | 0.57 | 0.43 | 1. |
| FACTCHECK $\mathcal{D}_{ca}$ | PHI3.5+DiscVBLLMLP$_{rw50}$ | VBLL | 0.98 | 0.34 | 0.98 | 0.33 | 0.98 | 0.34 | 0.98 | 0.33 | 0.98 | 0.67 |
| FACTCHECK | PHI3.5+DiscVBLLMLP$_{rw50}$ | VBLL | 0.35 | 0.11 | 0.99 | 0.32 | 0.90 | 0.04 | 0.82 | 0.39 | 0.83 | 0.43 |
| FACTCHECKSHUFFLED | PHI3.5+DiscVBLLMLP$_{rw50}$ | VBLL | 0.33 | 0.33 | R | 0. | 1. | 0.11 | 0. | 0.22 | 0.33 | 0.33 |
| FACTCHECK $\mathcal{D}_{ca}$ | PHI3.5+GenVBLLMLP | NO-REJECT | 0.90 | 0.50 | 0.93 | 0.50 | 0.93 | 0.49 | 0.91 | 0.51 | 0.92 | 1. |
| FACTCHECK | PHI3.5+GenVBLLMLP | NO-REJECT | 0.41 | 0.51 | 0.93 | 0.49 | 0.87 | 0.24 | 0.60 | 0.76 | 0.67 | 1. |
| FACTCHECKSHUFFLED | PHI3.5+GenVBLLMLP | NO-REJECT | 0.40 | 1. | R | 0. | 1. | 0.40 | 0. | 0.60 | 0.40 | 1. |
| FACTCHECK $\mathcal{D}_{ca}$ | PHI3.5+GenVBLLMLP | VBLL | 0.98 | 0.33 | 0.99 | 0.32 | 0.99 | 0.33 | 0.98 | 0.33 | 0.98 | 0.65 |
| FACTCHECK | PHI3.5+GenVBLLMLP | VBLL | 0.31 | 0.07 | 0.99 | 0.30 | 0.83 | 0.02 | 0.87 | 0.34 | 0.87 | 0.37 |
| FACTCHECKSHUFFLED | PHI3.5+GenVBLLMLP | VBLL | 0.20 | 0.22 | R | 0. | 1. | 0.04 | 0. | 0.18 | 0.20 | 0.22 |
| FACTCHECK $\mathcal{D}_{ca}$ | PHI3.5+GenVBLLMLP$_{rw50}$ | NO-REJECT | 0.91 | 0.50 | 0.93 | 0.50 | 0.93 | 0.49 | 0.91 | 0.51 | 0.92 | 1. |
| FACTCHECK | PHI3.5+GenVBLLMLP$_{rw50}$ | NO-REJECT | 0.41 | 0.51 | 0.93 | 0.49 | 0.87 | 0.24 | 0.60 | 0.76 | 0.67 | 1. |
| FACTCHECKSHUFFLED | PHI3.5+GenVBLLMLP$_{rw50}$ | NO-REJECT | 0.44 | 1. | R | 0. | 1. | 0.44 | 0. | 0.56 | 0.44 | 1. |
| FACTCHECK $\mathcal{D}_{ca}$ | PHI3.5+GenVBLLMLP$_{rw50}$ | VBLL | 0.98 | 0.33 | 0.99 | 0.29 | 0.99 | 0.33 | 0.98 | 0.29 | 0.99 | 0.62 |
| FACTCHECK | PHI3.5+GenVBLLMLP$_{rw50}$ | VBLL | 0.29 | 0.06 | 0.98 | 0.24 | 0.80 | 0.02 | 0.86 | 0.28 | 0.85 | 0.30 |
| FACTCHECKSHUFFLED | PHI3.5+GenVBLLMLP$_{rw50}$ | VBLL | 0.22 | 0.21 | R | 0. | 1. | 0.04 | 0. | 0.16 | 0.22 | 0.21 |
| FACTCHECK $\mathcal{D}_{ca}$ | MIXTRAL8x7B+DiscVBLLMLP | NO-REJECT | 0.91 | 0.50 | 0.93 | 0.50 | 0.93 | 0.49 | 0.91 | 0.51 | 0.92 | 1. |
| FACTCHECK | MIXTRAL8x7B+DiscVBLLMLP | NO-REJECT | 0.66 | 0.51 | 0.88 | 0.49 | 0.86 | 0.40 | 0.71 | 0.60 | 0.77 | 1. |
| FACTCHECKSHUFFLED | MIXTRAL8x7B+DiscVBLLMLP | NO-REJECT | 0.84 | 1. | R | 0. | 1. | 0.84 | 0. | 0.16 | 0.84 | 1. |
| FACTCHECK $\mathcal{D}_{ca}$ | MIXTRAL8x7B+DiscVBLLMLP | VBLL | 0.98 | 0.32 | 0.99 | 0.31 | 0.99 | 0.31 | 0.98 | 0.31 | 0.99 | 0.62 |
| FACTCHECK | MIXTRAL8x7B+DiscVBLLMLP | VBLL | 0.85 | 0.08 | 0.97 | 0.27 | 0.89 | 0.08 | 0.95 | 0.27 | 0.94 | 0.35 |
| FACTCHECKSHUFFLED | MIXTRAL8x7B+DiscVBLLMLP | VBLL | 0.91 | 0.14 | R | 0. | 1. | 0.13 | 0. | 0.01 | 0.91 | 0.14 |
| FACTCHECK $\mathcal{D}_{ca}$ | MIXTRAL8x7B+DiscVBLLMLP$_{rw50}$ | NO-REJECT | 0.90 | 0.50 | 0.94 | 0.50 | 0.94 | 0.48 | 0.90 | 0.52 | 0.92 | 1. |
| FACTCHECK | MIXTRAL8x7B+DiscVBLLMLP$_{rw50}$ | NO-REJECT | 0.62 | 0.51 | 0.89 | 0.49 | 0.86 | 0.37 | 0.69 | 0.63 | 0.75 | 1. |
| FACTCHECKSHUFFLED | MIXTRAL8x7B+DiscVBLLMLP$_{rw50}$ | NO-REJECT | 0.81 | 1. | R | 0. | 1. | 0.81 | 0. | 0.19 | 0.81 | 1. |
| FACTCHECK $\mathcal{D}_{ca}$ | MIXTRAL8x7B+DiscVBLLMLP$_{rw50}$ | VBLL | 0.98 | 0.33 | 0.99 | 0.34 | 0.99 | 0.32 | 0.98 | 0.34 | 0.99 | 0.67 |
| FACTCHECK | MIXTRAL8x7B+DiscVBLLMLP$_{rw50}$ | VBLL | 0.72 | 0.12 | 0.97 | 0.30 | 0.91 | 0.09 | 0.90 | 0.32 | 0.90 | 0.42 |
| FACTCHECKSHUFFLED | MIXTRAL8x7B+DiscVBLLMLP$_{rw50}$ | VBLL | 0.82 | 0.16 | R | 0. | 1. | 0.13 | 0. | 0.03 | 0.82 | 0.16 |
| FACTCHECK $\mathcal{D}_{ca}$ | MIXTRAL8x7B+GenVBLLMLP | NO-REJECT | 0.91 | 0.48 | 0.93 | 0.52 | 0.92 | 0.48 | 0.92 | 0.52 | 0.92 | 1. |
| FACTCHECK | MIXTRAL8x7B+GenVBLLMLP | NO-REJECT | 0.63 | 0.51 | 0.88 | 0.49 | 0.85 | 0.38 | 0.70 | 0.62 | 0.76 | 1. |
| FACTCHECKSHUFFLED | MIXTRAL8x7B+GenVBLLMLP | NO-REJECT | 0.73 | 1. | R | 0. | 1. | 0.73 | 0. | 0.27 | 0.73 | 1. |
| FACTCHECK $\mathcal{D}_{ca}$ | MIXTRAL8x7B+GenVBLLMLP | VBLL | 0.96 | 0.32 | 0.99 | 0.40 | 0.99 | 0.31 | 0.97 | 0.41 | 0.98 | 0.72 |
| FACTCHECK | MIXTRAL8x7B+GenVBLLMLP | VBLL | 0.58 | 0.13 | 0.99 | 0.33 | 0.95 | 0.08 | 0.85 | 0.38 | 0.87 | 0.47 |
| FACTCHECKSHUFFLED | MIXTRAL8x7B+GenVBLLMLP | VBLL | 0.54 | 0.11 | R | 0. | 1. | 0.06 | 0. | 0.05 | 0.54 | 0.11 |
| FACTCHECK $\mathcal{D}_{ca}$ | MIXTRAL8x7B+GenVBLLMLP$_{rw50}$ | NO-REJECT | 0.92 | 0.48 | 0.91 | 0.52 | 0.90 | 0.49 | 0.93 | 0.51 | 0.91 | 1. |
| FACTCHECK | MIXTRAL8x7B+GenVBLLMLP$_{rw50}$ | NO-REJECT | 0.67 | 0.51 | 0.89 | 0.49 | 0.87 | 0.40 | 0.72 | 0.60 | 0.78 | 1. |
| FACTCHECKSHUFFLED | MIXTRAL8x7B+GenVBLLMLP$_{rw50}$ | NO-REJECT | 0.69 | 1. | R | 0. | 1. | 0.69 | 0. | 0.31 | 0.69 | 1. |
| FACTCHECK $\mathcal{D}_{ca}$ | MIXTRAL8x7B+GenVBLLMLP$_{rw50}$ | VBLL | 0.97 | 0.38 | 0.98 | 0.40 | 0.98 | 0.37 | 0.97 | 0.41 | 0.97 | 0.78 |
| FACTCHECK | MIXTRAL8x7B+GenVBLLMLP$_{rw50}$ | VBLL | 0.67 | 0.19 | 0.95 | 0.35 | 0.89 | 0.14 | 0.84 | 0.39 | 0.85 | 0.53 |
| FACTCHECKSHUFFLED | MIXTRAL8x7B+GenVBLLMLP$_{rw50}$ | VBLL | 0.67 | 0.15 | R | 0. | 1. | 0.10 | 0. | 0.05 | 0.67 | 0.15 |

