# OpenReview forum: "Similarity-Distance-Magnitude Activations"
_TMLR — Rejected by TMLR_

### Review · Reviewer_UjjG · 2025-06-29

**Summary Of Contributions:**

This paper introduces a SDM activation function as an alternative to the softmax activation function with three components: Similarity (S) measures how well an input matches its nearest neighbours in feature space, Distance (D) calculates how far it is from the calibration samples in feature space, and Magnitude (M) which behave as raw logits. These components are combined to adjust the model’s output probabilities, aiming to improve uncertainty estimation. To calibrate the resulting confidence scores, the authors apply class-conditional empirical cumulative distribution functions and a rescaling transform based on held-out calibration data. The SDM framework is extended to large language models via SDM Networks, which include a verification layer and regularization strategies to preserve previously learned knowledge.

**Audience:**

Yes

**Broader Impact Concerns:**

N.A.

**Claims And Evidence:**

No

**Requested Changes:**

# Requested Changes

1.. Many acronyms are not defined. Although some of them, such as CDF, can be reasonably conjectured, the rest, such as HCI and DKW, are mysterious. It would be clearer to state the explicit terms before using their acronyms.

2. The loss function, as defined in Eq. (8), should also be a function of the calibration set ($\mathcal{D}_{\text{ca}}$), which is also involved in the calculation.

3. SDM Activation (Sec. 4.1.5, Eq. (7)): The selection of the scalar (2) lacks justification. Please provide more details, such as whether this is a result of hyperparameter tuning.

4. Address the concerns on the proposed distance $d$ in weaknesses 2-5.

5. Address the concern on the proposed similarity $q$ in weakness 1.

6. Address the concern on the proposed SDM activation in weakness 6.

**Strengths And Weaknesses:**

# Strengths
1. Propose an SDM activation function as an alternative to the standard softmax activation function for the classification models, taking into account the L2 distance in the feature space.

2. Utilise empirical CDFs over a calibration set to transform the model prediction into epistemic uncertainty.

# Weaknesses

1. The proposed similarity, as defined in Eq. (5), it is highly dependent on the compactness of class-wise features extracted by the base model (parameterised by $\theta$) due to the requirement for consecutive matches. This is encoded in the third indicator function which requires the ($i-1$)-nearest features to be class consistent. However, such a property is not guaranteed.

2. The statement (1st paragraph in Sec. 4.1.3) justifies the proposed distance function, as defined in Eq. (6), with difficulty working with $d_{\text{nearest}}$ directly on the train set. However, the proposed distance function still requires $d_{\text{nearest}}$, which demands comparing the L2-distance between the selected data point and each of the train data points. Also, it still depends on the input and output dimensions and the structure of the adapter $g$. This totally contradicts the justification. In addition, the empirical CDFs in the proposed distance method requires computing the L2-distance between the selected data point and each of the data point in the calibration set  This also depends on the adaptor $g$, which requires additional computational overhead.

3. The distance function (Eq. (6)) computes the smallest difference between a scalar (1) and the empirical class-wise CDF (constructed with the empirical L2 distance between the selected data point and each of the data point in the calibration set) of $d_{\text{nearest}}$. Unless the calibration data points are locally distributed around the selected data point in the feature space, which is highly unlikely, the distance function should almost always return a value of approximately 1.

4. The statement in Sect. 4.1.5 suggests that higher distance values ($d$) should encourage a sharper predictive distribution. However, a higher distance value indicates that the selected point is actually isolated from the calibration set. This contradicts the quoted statement suggesting that the predictive distribution should be uniform for out-of-distribution data ($d = 0$).
> Also note that when $d_{\text{nearest}}$ exceeds the largest distance observed in the labeled data, $d = 0$ and the output distribution is uniform, reflecting a maximally high (i.e., out-of-distribution) epistemic uncertainty estimate.

5. The statement in Sec. 4.1.5 corresponds $d = 0$ to out-of-distribution data. However, this can only happen if all data points in the calibration set, despite their groundtruth classes, are locally distributed around the selected data point in the feature space. This happens only when there is a severe covariate shift between the train data and the calibration data, and the base model also happens to map all calibration data points to the local neighbourhood of the selected data point in the feature space. In addition, $d = 1$ should also correspond to the out-of-distribution data, as it indicates that the selected data point is isolated from the calibration data in the feature space.

6. The proposed SDM activation (Sec. 4.1.6) can only backpropagate the gradients through the magnitude $z'$. The backpropagations through similarity $q$ and distance $d$ are prohibited as the indicator function, and the empirical CDFs are not differentiable. Thus, optimising the loss function defined in Eq. (8) would only optimise the magnitude values, which would behave similarly to the logits in the softmax function.

7. The experimental results are not convincing. In most of the cases, the proposed methods only trade acceptance rate for accuracy. Especially on the Sentiment OOD and Factcheck datasets, the proposed method could even result in total rejection of the predictions in some cases.

---

### Review · Reviewer_AZTg · 2025-07-11

**Summary Of Contributions:**

This work proposes a novel strategy for providing more calibrated neural network models. It does so by introducing a novel activation function called the SDM activation that is inspired by the usual softmax activation function. The SDM activation utilises three quantities dubbed "similarity", "distance" and "magnitude". These quantities are computed from certain statistical intuitions that should intuitively encode signals of epistemic incertainty. Additionally, further refinements of the SDM function are introduced to also encode signals of effective sample sizes by rescaling as well as cutting of predictions in regions below a desired confidence threshold. All of these changes have the aim of trying to enforce a notion of calibration that the authors introduced, called index-conditional calibration. Finally, the step is made from SDM activation functions to SDM networks, which augment traditional LLMs with an exemplar adaptor and replaces all softmax activations with SDM activations. Both SDM activations and SDM networks are tested on three benchmarks by measuring their class-conditional, prediction-conditional or regular accuracies under an assumed 0.95 threshold.

**Audience:**

Yes

**Claims And Evidence:**

No

**Requested Changes:**

Please find my precise request and questions above intertwined with the strengths and weaknesses.

**Strengths And Weaknesses:**

#### Strengths
The overall goal of the work is important; neural networks remain at the forefront of AI developments, but struggle with retaining accuracy in out-of-distribution settings or more generally "knowing what they don't know". Some of the provided empirical results do show that the SDM networks can simply "refuse" to output anything if their computed notion of confidence drops below a certain threshold.

The perspective on calibration presented in the work is quite different from usual approaches. That is, the authors try to include information about the uncertainty on certain data instances with respect to the training data or a held-out calibration set in the activation function. Other approaches more generally take a post-hoc, loss-based or conformal perspective.
#### Weaknesses
Unfortunately, the paper does have quite significant weaknesses in its current form, please find specific issues below.

**The exposition of the paper is convoluted, hard to follow and disconnected.** Starting at the introduction, it is already not clear what the overall goal of the paper is. Many strong claims are made, e.g. "we address the neural network robustness problem" or "We [...] further address the complementary HCI problem", that are not supported. The general goal seems to simply be to introduce a novel perspective and method for neural network calibration, i.e. adding mechanisms to make neural predictions aware of "what they don't know". Instead of focussing on this clearly important goal, the authors make the earlier stronger claims and add more confusing goals of LLM verification. Next, much of the story of the text itself is quite disconnected. For instance, much of Section 4 introduces vague problems and subsequently posits certain solutions without it being clear that the solution solves the problem. In short, there is a distinct disconnect between stated vague problem, e.g. "SDM [...] only has an indirect, relative notion of effective sample size" (start Section 4.2.1), and given solution, e.g. "We rescale $q$ to take into account these distributional statistics". Finally, it is very hard to grasp what sort of notion of *verifiability* is actually used throughout the paper. It seems the authors are simply training a separate neural network for a binary notion of verification, but that does not give any real guarantees due to being at the mercy of how well the optimisation goes. In a strict sense, you are not verifying the output, only giving an indication of verification. In general, these point show how much of the paper is lacking explicit statements of problems, goals and solution.

**The paper proposes many steps of transforms or other mechanism without providing meaningful evidence of why each of the steps are necessary or why the steps yield the desired result.** Focusing on Section 4, there is much written about different transforms that are *supposed* to solve certain weaknesses that the original SDM might have. However, I am lacking an explicit and precise explanation of *why* these problems would arise and *why* the given solution solves the problem. Right now, Section 4 mostly looks like a chaotic technical report where the authors report a number of intuitive mechanisms that together are somehow supposed to give better calibration results. No theoretical evidence is given that some of the proposed mechanisms, e.g. the ones in Sections 4.2.1 or 4.2.4, indeed solve the problem they were set out to solve. Due to the lack of a proper ablation study in the empirical section, there also does not seem to be any experimental evidence for the necessity of all components either.

**The experimental section seems to lack a proper ablation study and seems somewhat inconsistent.** As stated in the previous paragraph, one major oversight of the paper seems to be a proper ablation study that confirms that all proposed mechanisms of Section 4 are solving the right problems. That is, the authors claim there is a problem such as accounting for uncertainty in data splitting (Section 4.2.4), propose a solution, but never verify that solution. To be scientific, one needs a metric to measure if the solution solves the problem, which is not happening in many cases here. Even in the overall picture, only either the base SDM activation function is used or the full SDM network with all transforms. To have a complete ablation study, the authors should also incrementally add components to analyse which transforms have what kind of effect. Apart from these missing ablations and analyses, I don't fully understand why some of the baselines change between experiments. Specifically, why can you not use the estimators for the experiment in Section 5.1 for the experiment in Section 5.2 as well, and vice versa? Why did you also not consider loss function-based [1, 2] or regularisation-based [3] calibration methods?

**Mathematics in the paper are at times handwavy or even wrong.** There are a couple of occasions where mathematics are not precise. For instance, the general notation of probability is often confused with notation for probability mass function. As a very specific example, the authors use the notation $p(y = \hat{y} \mid \hat{y}, \mathcal{E}(x) = \alpha)$ in Definition 4.1 and $p(y = \hat{y} \mid y, , \mathcal{E}(x) = \alpha)$, but what do these mean? What are the random variables here and what values of the random variables are under consideration? You probably want to express the probability that the random variable $Y$ (label) takes the value $\hat{y}$ (predicted value) given that the estimator predicts a probability of $\alpha$, which would be correctly denoted by $\mathbb{P}(Y = \hat{y} \mid \mathcal{E}(x) = \alpha)$. Another example appears in Section 4.2.4 where it is said to "estimate each of these processes as Cauchy distributions". However, it is unclear what "these processes" are, i.e. what exactly are the random variables here that are supposed to follow a Cauchy distribution? Finally, in Section 4.2.3, the authors use the concept of probability simplex, which is the set of all possible finite distributions over a certain number of outcomes. However, they state that the output of the softmax is a probability simplex, while it is only an element of the simplex.

#### Verdict

The presentation of the paper leaves much to be desired, looking more akin to a technical report than a scientific publication. Methods are proposed to abstract and vaguely stated problems without them being verified theoretically or empirically. The authors should perform at least and ablation study to find out at least experimentally if each of the components do what they should do. Apart from that, the presentation also seems quite disconnected and is hard to follow. I would highly recommend the authors to loosen their overly strong initial claims and focus on the real problem that is being solved here: calibration of neural networks and the benefit that comes with that. There are plenty of interesting points in the paper, especially the part where the authors propose a search strategy to determine when the model can actually predict something meaningful (Section 4.2.3). In its current form, I am partial to rejection, but I can be convinced for a major revision if the authors are willing to address all of my raised issues.

[1] Mukhoti, Jishnu, et al. "Calibrating deep neural networks using focal loss." Advances in neural information processing systems 33 (2020): 15288-15299.

[2] Karandikar, Archit, et al. "Soft calibration objectives for neural networks." Advances in Neural Information Processing Systems 34 (2021): 29768-29779.

[3] Guo, Chuan, et al. "On calibration of modern neural networks." International conference on machine learning. PMLR, 2017.

---

### Review · Reviewer_Cctf · 2025-09-02

**Summary Of Contributions:**

This paper presents a new output-activation function called SDM by modifying the standard softmax function: in my opinion, the key modification is that the standard exp(.) function is replaced by a pow(k,.) where k is learnable. Then, the paper proceeds to demonstrate the effectiveness of this output-activation for calibration and robustness use-cases. This paper combines several techniques to natively incorporate the uncertainty-aware prediction.

**Audience:**

Yes

**Broader Impact Concerns:**

NA.

**Claims And Evidence:**

Yes

**Requested Changes:**

* The main request is to refactor the paper for improved readability. I understand that having detailed pseudocodes and derivation is very valuable, but I hope the authors can rewrite from the viewpoint of a reader unfamiliar with this work. If published in current form, my opinion is that this paper would not achieve the desired impact due to its presentation format.

* I found the results from table 1/4/5 are quite encouraging --- an associated qualitative analysis would be very valuable.

**Strengths And Weaknesses:**

Strengths:
* This paper is extremely well written and very detailed. (this is also ironically the main weakness!)
* The proposal is evaluated on a variety of tasks and compared against relevant baselines.


Weaknesses:
* The major weakness is that the paper requires a lot of time to parse and I found myself getting lost in the middle. It is in the authors' own best interests to refactor this paper for ease of readability.
* Also, it appears that the training-dataset needs to be held in memory during evaluation. Is this a fair understanding?
* The paper is also missing an ablation/design-consideration when introducing various components (e.g., rescaling/cdf/etc). Is there a simpler way to achieve the same results?

---

> ### Author Response · Authors · 2025-09-08
> **Revision**
>
> Thanks for the suggestions. The revision, to be posted this week, will be a significantly shorter abridged version focused more specifically on the [behavior of the] SDM activation function.
>
> We will post the PDF shortly, but here is the abstract in the interim:
>
> Similarity-Distance-Magnitude Activations
>
> We introduce a more robust and interpretable formulation of the standard softmax activation function commonly used with neural networks by adding Similarity (i.e., correctly predicted depth-matches into training)-awareness and Distance-to-training-distribution-awareness to the existing output Magnitude (i.e., decision-boundary)-awareness. When used as the final-layer activation with language models, the resulting Similarity-Distance-Magnitude (SDM) activation function is more robust than the softmax function to co-variate shifts and out-of-distribution inputs in high-probability regions. This makes it preferable for selective classification settings, even when considering post-hoc calibration methods over the softmax.

---

### Author Response · Authors · 2025-11-14
**Summary**

In our controlled examination of representative Frequentist, Bayesian, and empirically motivated estimators, our proposed SDM estimator (i.e., the SDM activation function combined with the selection criteria of Eq. 9) is the only estimator that remains well-calibrated in high-probability regions across the in-distribution, co-variate shifted, and out-of-distribution data. In this way, the SDM estimator provides a principled, data-driven substrate for approaching selective classification, calibration, and out-of-distribution detection with language models.

---

### Author Response · Authors · 2025-11-23
**Summary (1 of 2)**

In summary, our work is a 12 page research paper addressing the following well-defined calibration problem, rather than a technical report about a product, which contrasts with the AE's comment. In short:

> Given a held-out test set, can we reliably detect the set of points for which the class-conditional accuracy is at least $\alpha$ and the prediction-conditional accuracy is at least $\alpha$? We seek a conservative notion of calibration in which we prefer a smaller non-rejected set over falling under the specified $\alpha$ value, but given two estimators that are calibrated in the aforementioned sense, we prefer that which rejects fewer points.

This reflects the language model use-case of using the outputs in the well-calibrated, high-probability region as automated, or semi-automated, predictions in the decision pipeline, with output outside that specified acceptance region being sent to additional models, retrieval, human adjudication, etc.

We examine this question in a carefully controlled series of experiments, examining recent Frequentist (e.g., conformal), Bayesian (VBLL), and empirically-motivated approaches (temperature scaling, thresholding the softmax, etc.). Unlike typical calibration papers, we also examine the behavior over co-variate shifted and out-of-distribution data. (Contrast that with the comments by reviewer AZTg regarding missing comparisons. It is unclear specifically what additional comparisons are still sought in the revised paper. Of note is that reviewer AZTg suggested we had not compared to [1], but indeed we had: That is temperature scaling. Ironically, in the AE's own work, [2], which we compare to, the comparison in the text classification experiments is only to an MLP baseline, representing an absence of a post-hoc calibration method.)

As noted some months ago to the reviewers, we are of the position that typical calibration papers make a much stronger (and unrealistic) implicit claim: That marginal (cf., [approximately] conditional) calibration is sufficient and that evaluation over in-distribution data is sufficiently informative for comparisons. For example, the 4 document classification tasks in [1] are all in-distribution.

By our reading, the language used in our paper is quite modest and measured, in contrast to the under-specified claims of over-claiming by one of the reviewers. Contrast that with, for example:

- "Our most important discovery is the surprising effectiveness of temperature scaling despite its remarkable simplicity....Nevertheless, simple techniques can effectively remedy the miscalibration phenomenon in neural networks" [1]. That would suggest that the problem was done in 2017. Unfortunately, in reality, temperature scaling works poorly in the presence of co-variate shifts, as we demonstrate.

- "These findings validate the VBLL models' potential for integration with large-scale modern language models for diverse applications, particularly in sentiment classification tasks" [2]. However, the baseline in those experiments is an MLP, not a calibration method. As we showed in Appendix A.4, the VBLL estimator is not well-calibrated in high-probability regions over the co-variate shifted data with the examined models, which are representative open-weight, decoder-only LMs.

The questions by reviewer UjjG were answered in the lengthy 2 part response on July 1, to which the reviewer did not respond. We briefly summarize those points here for reference in a public comment.

Our approach expands on an empirical pattern first observed in [3]: Exemplar adaptors, 1-D CNN adaptors (with a final linear layer) over the frozen hidden states of a network, induce distilled, compressed representations of an underlying network's representation space conditional on its predictions. We can use these representations to approximate the predictions of the underlying model. It was observed in [3] that when the approximations diverge from the predictions of the underlying model, the inputs tend to be from the regions of the distribution over which the underlying model is itself unreliable. This suggests that it may be possible to use these signals to model the epistemic uncertainty, which is the subject of the current work.

The work of [3] requires training an adaptor layer, and then separately training an approximation of that adaptor layer with a KNN model that aims to minimize the prediction discrepancies between the predictions from the adaptor layer and that of the KNN model. In our work, we simplify that pipeline by training the adaptor directly with the SDM activation layer, eschewing a separate KNN approximation.

(continued)

---

> ### Author Response · Authors · 2025-11-23
> **Summary (2 of 2)**
>
> (continued from above)
>
> The approximations in [3] are trained with a BCE loss, where the stopping criteria is the minimum count of prediction discrepancies between the original model and the approximation. If the immediately preceding epoch does not yield the minimal count of discrepancies among all epochs, the loss is subsequently only calculated for the instances with prediction discrepancies until a new minimum discrepancy count is found, or the maximum number of epochs is reached. This has a regularizing effect: There is signal in the magnitude of the KNN output, so training aims to optimize in the direction of minimizing the residuals; however, the aforementioned masking approach also seeks to avoid over-fitting to the magnitude of the outliers. Instead, in our work, instance-wise q and d serve as the regularizers during training, reflecting the error distribution modeled by the SDM output as used at test-time. It is not true that these regularization terms have no effect on training. Separately, as noted in Footnote 4, by design, we are not back-propagating through a bi- and/or cross-encoded search graph, which is a different architecture and orthogonal use-case for dense matching (e.g., semantic matching needed when training a neural information retrieval system with contrastive losses).
>
> It is not true that the approach is at the "mercy of how well the optimisation goes" (reviewer AZTg) in the sense of an approach like temperature scaling, since Algorithm 1 to find $q'_\rm{min}$ runs after the parameters have been fixed. Given the fixed parameters, it estimates the partitioning of the distribution based on the eCDFs over the output.
>
> Regarding the behavior of $d$, the distance quantiles: $d=1$ corresponds to $d_\rm{nearest}=0$, which occurs when the test instance has an exact match in the training set. $d=0$ occurs when the test instance has a $d_\rm{nearest}$ value that exceeds the observed $d_\rm{nearest}$ values over the calibration set. In practice with in-distribution data, $d$ is not "almost always" 0 or 1. Note that each test instance has one value of $d_\rm{nearest}$, and each $eCDF_\rm{ca}$ is a per-class quantile function. (That is, $d_\rm{nearest}$ is used as an index into the class-wise distribution of $d_\rm{nearest}$ distances of each calibration point to the training set.) The conservative approach of taking the min across classes is purposeful for robustness against the relative skew of the distribution of $d_\rm{nearest}$ across classes.
>
> The offset constant should be some small value > 1 since $q \in \lbrace 0, \ldots, \vert D_\rm{tr} \vert \rbrace $ and we want $(\rm{offset} + q)^{d\cdot z'_i}$ to be increasing in each of its constituent values, ceteris paribus. The important point is to be consistent, provided it is some reasonably small value to avoid saturating the differences across q; however, there is no particular reason to set it to a different value than our convention of 2, which is simply the smallest integer fulfilling the aforementioned requirement.
>
> There was some confusion in interpreting the results, conflating calibration with offsetting of thresholds to trade acceptance for accuracy. That is not the calibration setting. The calibration setting is generally more challenging and needed for real-world applications. Given a particular $\alpha$, we seek for the admitted (non-rejected) points to have class-conditional and prediction-conditional accuracy at least $\alpha$. Our stated goal with calibration in our work is the selective classification setting where we seek to reliably detect the high probability region. The behavior on SentimentOOD and the shuffled datasets is as desired, ceteris paribus. With other calibration methods, we would risk introducing silent errors into our LM pipelines, as the alternative calibration methods tend not to be well-calibrated over such shifts, resulting in over-confident estimates. Importantly, at the same time, the SDM estimator is not vacuously conservative, as shown with the Sentiment task and the results in Table 3, for example.
>
> More broadly, this is critical missing behavior for real-world deployments of LMs, because in the absence of this behavior, it is challenging to build decision pipelines that can reliably "know what they do and do not know". We seek LMs that can fail gracefully, rather than unexpectedly go-off-the-rails.
>
> [1] Chuan Guo, Geoff Pleiss, Yu Sun, and Kilian Q. Weinberger. On Calibration of Modern Neural Networks. *ICML*, 2017.
>
> [2] James Harrison, John Willes, and Jasper Snoek. Variational Bayesian Last Layers. *ICLR*, 2024.
>
> [3] Allen Schmaltz. Detecting Local Insights from Global Labels: Supervised & Zero-Shot Sequence Labeling via a Convolutional Decomposition. *Computational Linguistics*, 2021. Presented at *EMNLP*, 2021.

---

> > ### Comment · Action_Editor_xYHH · 2025-12-01
> > **Re**
> >
> > Dear author, I appreciate your comment and sympathize.  I apologize if my meta-review was taken in offense, it was not meant to.  Unfortunately, the reality is that only one reviewer was willing to accept the paper and their decision wasn't a strong endorsement: "Compared to the original submission, the current submission is certainly more readable. However, the authors do not answer the key questions raised by me or the other reviewers (perhaps, they're not visible to me?). It does seem to reasonably demonstrate the claims but I will not fight for acceptance.".  It seems they still had lingering concerns.  Certainly, the reviewers' consensus was against acceptance.
> >
> > Upon reading the paper and reviews some more, I understand where the reviewers are coming from.  It feels as if there is a disconnect between the intro and the presentation of the method.  For example, I agree that I do not quite see how the method is maximizing the separation of aleatoric and epistemic uncertainty "decomposing the sources of the latter in a manner that is interpretable and actionable".  The intro reads as if to present the work as solving a foundational problem with uncertainty quantification in deep learning "A foundational piece is evidently missing from the picture."   In my opinion, the introduction sets the bar extremely high for the method (and for TMLR it is important that the claims are supported by the paper).  However, as far as I can tell, there isn't a theoretical reason that the method should be robust, well-calibrated, etc.  The method comes across as a recipe for a series of heuristics (each of which seems intuitively reasonable), but it's hard to follow why this particular setup is addressing the desiderata from the intro.  In Section 3.2 it seems like the justification is coming (this is using conformal prediction to give guarantees), but it's difficult to follow the writing given caveats and special casing.
> >
> > Perhaps some theoretical justification or even some more intuition as to why this method should work would go a long way to making the paper stronger.  It's possible that this is largely a clarity or presentation issue with the writing.

---

### Decision · Action_Editor_xYHH · 2025-11-03

**Recommendation:** Reject

**Additional Comments:**

Two of the reviewers voted for reject, while one voted lean accept.  The reviewer who voted "lean accept" unfortunately was unwilling to argue strongly for acceptance.  Their concerns echoed those of the other two reviewers - that the paper lacked the technical detail, experimental results and ablations to convince them that the claims about the method were justified.  At times, it reads more like a tech report describing a product than an ML paper.  The reviews are quite thorough and included a variety of comments on how to improve the paper.  I would recommend summarizing the tables that take up most of pages 9, 10 & 11 and putting the full tables in an appendix.  Then you could use the reclaimed space to provide some more detail regarding the claims (e.g. about aleatoric & epistemic uncertainty) and add ablations, etc.

Note that the bar for TMLR is not really whether or not the presented work is state-of-the-art, but instead whether the technical contribution is correct, relevant and sufficiently detailed for publication, and that whatever claims are made by the authors are substantiated in the text and experiments.  It seems like this paper emphasized the state-of-the-art claims but did not provide sufficient detail and empirical evidence to substantiate those claims.  If this paper were to be resubmitted with major revisions, I'd recommend providing more detail, making the exposition clearer and toning down the claims (or otherwise significantly strengthening the experiments).

**Audience:**

Yes

**Audience Explanation:**

Uncertainty quantification and robustness to out-of-distribution inputs remain two important topics in machine learning.  This paper claims to develop a simple but highly effective method for quantifying model uncertainty - and thus could be quite impactful and of interest to the community.  So, yes it does seem like it is of interest.

**Claims And Evidence:**

No

**Claims Explanation:**

One of the reviewers found the claims were supported by evidence while two argued they did not.  Both reviewers commented that they found the experiments to not justify the claims made in the paper.  They found some results to just not be as strong as claimed and they felt the paper was lacking ablation studies necessary to make claims about the effectiveness of the transforms.  A common concern was that the reviewers found the paper to read more like a technical report or advertisement of the method rather than a full research paper with the necessary technical details, ablations and justifications.  Therefore, the paper doesn't really meet the criteria of being supported by "accurate, convincing and clear" evidence.

**Resubmission Of Major Revision:**

The authors may consider submitting a major revision at a later time.